# THE MARKOVIAN THINKER: ARCHITECTURE-AGNOSTIC LINEAR SCALING OF REASONING

**Milad Aghajohari**[1,*]**, Kamran Chitsaz**[1,6,*]**, Amirhossein Kazemnejad**[1,*]**,**
Sarath Chandar[1,5,6,7], Alessandro Sordoni[1,2,†], Aaron Courville[1,5,8,†],
Siva Reddy[1,3,4,5,†]
[1]Mila    [2]Microsoft Research    [3]McGill University    [4]ServiceNow Research
[5]Canada CIFAR AI Chair    [6]Chandar Research Lab    [7]Polytechnique Montréal
[8]Université de Montréal  [*]Equal contribution    [†]Equal advising

## ABSTRACT

Reasoning LLMs suffer from quadratic compute growth as their context length increases, making reinforcement learning with verifiable rewards (RLVR) and test-time scaling prohibitively expensive. Prior work has tried to lighten the computational burden by shortening reasoning traces through pruning, summarization, or multi-stage training, but these methods remain bound to quadratic costs. We introduce Delethink, a thinking algorithm that realizes the Markovian Thinking Paradigm. Instead of producing one long monolithic reasoning trace, Delethink thinks in a sequence of chunks, the Delethink trace. Each chunk continues reasoning by referring only to a fixed number of prior tokens, which functions as a Markovian state sufficient for progressing reasoning, while deleting the rest. This preserves continuity without carrying the quadratic baggage. As a result, compute scales linearly and peak memory remains constant. In experiments, we show that Delethink can be applied directly to off-the-shelf reasoning models ranging from 1.5B to 30B parameters, with no loss in performance. Extended reasoning becomes possible under fixed memory and linear compute, while enabling efficient RL training on new tasks. On the DeepScaleR dataset, Delethink trains R1DistillQwen1.5B to the same benchmark performance as a standard long chain-of-thought (LongCoT) approach, where both models generate up to 24k thinking tokens. The difference is efficiency. Delethink reasons 40% faster with 70% less memory footprint. By decoupling reasoning length from context length, the Markovian Thinking paradigm opens the door to next-generation reasoning LLMs that can scale to millions of tokens with linear compute and constant memory.

## 1 INTRODUCTION

Reasoning LLMs answer questions by generating a long chain of thought (LongCoT) before giving a final answer, a method that has significantly advanced performance on many tasks and benchmarks (Jaech et al., 2024; Guo et al., 2025; OpenAI, 2025). Scaling these "thinking tokens" continues to push capability (Agarwal et al., 2025). When thinking tokens grow linearly the training and inference cost grows quadratically due to the quadratic complexity of self-attention, making LongCoT prohibitively expensive. For instance, a forward pass through a 1M-token context requires over a thousand times more compute than one with a 32K-token context. Moreover, throughput is inversely proportional to context length, reducing token/s at 1M context by over $30\times$ vs. 32k (Ao et al., 2025).

Prior work focused on cutting reasoning length to reduce the computational costs. During inference (Yan et al., 2025) iteratively summarizes past reasoning, (Lin et al., 2025) drops irrelevant or redundant tokens, and others distill shorter thinking styles with skipped steps (Xia et al., 2025; Liu et al., 2024a; Cheng & Van Durme, 2024; Han et al., 2024; Luo et al., 2025a). During RL training, Luo et al. (2025b) explored multi-stage training which limits the fraction of training that involves long reasoning traces and others implicitly reward shorter lengths among correct solutions (Aggarwal & Welleck, 2025; Shen et al., 2025; Li et al., 2025; Hou et al., 2025). Instead of cutting quadratic cost by cutting thinking, we want our algorithm to think more with self-attention with linear compute.

**LongCoT trace**

LongCoT reasoning (context grows linearly with more thinking tokens)

**Delethink trace** *(markovian thinking)*

Chunk 1   Chunk 2   Chunk $K - 1$   Chunk $K$

(context size remains constant throughout)

prompt    response

Figure 1: **Delethink Inference:** LongCoT generates a single long chain-of-thought. Delethink reasons in a sequence of short chunks, where each chunk conditions only on the original prompt and a small suffix of the previous chunk.

In this paper, we investigate if LLMs can reason effectively while retaining only a fixed number of recent thinking tokens and ignoring the rest. We call such a model, a *Markovian Thinker*, since the next reasoning steps depend only on the last chunk of thought, rather than the entire thought history. This avoids the quadratic cost while enabling the model to scale its thoughts with linear compute.

Current practice for training and scaling test-time compute in reasoning LLMs relies on long chain-of-thought (*LongCoT*) training, which incurs quadratic cost in context length. We introduce a simple novel algorithm, *Delethink*, which can be readily applied to off-the-shelf reasoning LLMs to make them Markovian thinkers, thereby scaling compute linearly with the number of thinking tokens. As shown in Figure 1, Delethink produces a sequence of short chunks; each chunk conditions on (i) the original prompt and (ii) a small suffix of the previous chunk, which together act as a sufficient (Markovian) state.

Surprisingly, *Delethink* is highly effective. We show that only carrying the last few thousand thinking tokens across chunks enables LLMs to continue their reasoning while matching or surpassing their LongCoT performance, even if the latter mirrors their training regime. For instance, Delethink inference boosts the performance of R1DistillQwen1.5B on AIME'24 by $9\%$ while extending its thinking to 128k tokens and simultaneously reduces costs in terms of both FLOPs and KV cache requirements. This happens while standard LongCoT inference rarely escapes the train-time thinking length. This is done in a zero-shot fashion, with neither prompting nor training. We observe the same pattern across open-source[1] reasoning models from 1.5B to 30B parameters and across benchmarks spanning PhD-level questions, coding tasks, and math competitions (Section 3.1).

We show that Delethink not only enables an LLM to function effectively as a Markovian thinker at inference time, but can also be used to train native Markovian thinkers, thereby making the entire reinforcement learning process significantly more compute-efficient than LongCoT. We trained R1DistillQwen1.5B (Guo et al., 2025) on the DeepScaleR dataset (Luo et al., 2025b), where both methods could think up to 24k. While the Delethink-trained model performs competitively to the LongCoT model, it generates tokens $40\%$ faster and reduces KV cache footprint by $70\%$. We empirically validate that training a model with an average thinking length of 96K tokens would require 27 H100-months using LongCoT and only 7 H100-months using Delethink (Figure 7).

We summarize our contributions:

- In this work, we show a simple method to build a Markovian thinker, that scales compute linearly and keeps memory constant during both training and inference.

---

[1]Evaluating this effect requires access to the model's thinking tokens, which precludes tests on closed-source models.

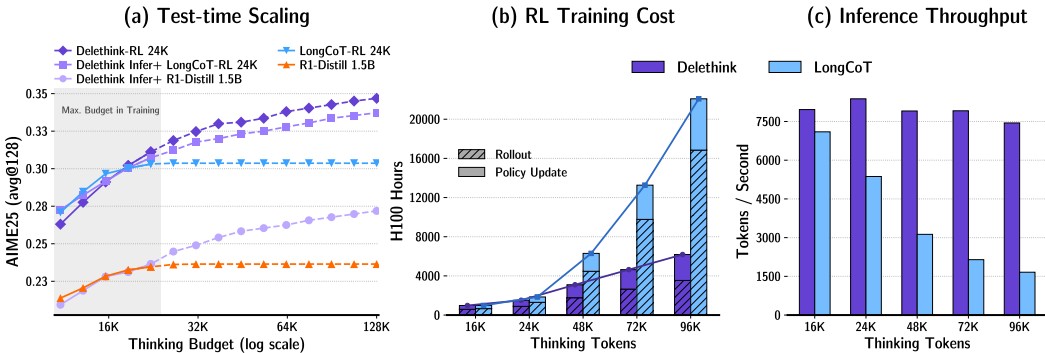

Figure 2: (**a**) A Delethink-trained checkpoint matches LongCoT's accuracy while scaling better at test time and using less compute. Delethink Inference also enables the LongCoT checkpoint and the original model to scale. (**b**) Training cost vs. average sequence length: quadratic for LongCoT and linear for Delethink, as predicted. (**c**) Generation throughput in sglang (Zheng et al., 2023) falls with context length; Delethink keeps context fixed, whereas LongCoT grows it linearly.

- We introduce Delethink Inference that enables off-the-shelf LLMs to be zero-shot Markovian thinkers.
- We introduce Delethink Training, that trains these Markovian thinkers through RL. We show it matches the performance of the standard long chain-of-thought in much less compute.

## 2    RELATED WORKS

**Efficient Thinking** Some works seek efficiency by shortening reasoning traces. Some distill traces with skipped steps or tokens (Liu et al., 2024a; Xia et al., 2025). Others control length by early exits (Ding et al., 2024), adjusting the budget to problem complexity (Han et al., 2024), or steering activations toward brevity (Zhao et al., 2025b). Structured prompting and collaboration help reduce tokens: CoThinking outlines then reasons (Fan et al., 2025). (Cheng & Van Durme, 2024) thinks in shorter trace of contemplative tokens. RL approaches include stabilizing GRPO to avoid wasted length (Liu et al., 2025), and training models to prune or exit early while maintaining accuracy (Luo et al., 2025a; Dai et al., 2025; Zhao et al., 2025a). Delethink is orthogonal, seeking not fewer tokens for the same performance but longer traces for better performance.

**Making Room for Extra Thinking** Prior work has tried to shorten reasoning traces at inference time. Lin et al. (2025) drops irrelevant tokens and redundant steps using a judge LLM. Xiao et al. (2025) prunes low-value segments by perplexity. Yan et al. (2025) iteratively summarizes current reasoning so later steps continue depending only on the summary. InftyThink's style is fixed by design, distilled from a hand-crafted dataset that locks the model into one pattern. Delethink, in contrast, learns through RL.

**Linear Architectures** Sliding or sparse attention (Beltagy et al., 2020; Zaheer et al., 2020) and kernel-based linear attention and low-rank linear attention (Katharopoulos et al., 2020; Choromanski et al., 2021; Wang et al., 2020) aim to avoid all-pairs interactions, the quadratic component of attention, by approximation or sparsity. Mamba architecture replaces self-attention with state-space models (Gu et al., 2021; Gu & Dao, 2023; Dao & Gu, 2024), achieving constant-memory, linear-time generation via recurrent state updates. Hybrid systems interleave attention with alternatives which scale asymptotically quadratic (Lieber et al., 2024; AI21 Labs, 2024). Our approach is orthogonal to these efforts that linearize attention or replace it. We keep the underlying model unchanged without relying on attention approximations or architecture swaps.

**KV Eviction and Compression** To cut inference memory without retraining, eviction policies keep only the most useful past tokens. Zhang et al. (2023); Yang et al. (2024) select important tokens based on their estimated contribution to attention, while Xiao et al. (2024) preserves a small set of attention-sink tokens to stabilize quality under sliding windows. Compression approaches shrink each retained token's footprint via quantization to sub-4-bit while maintaining accuracy (Hooper

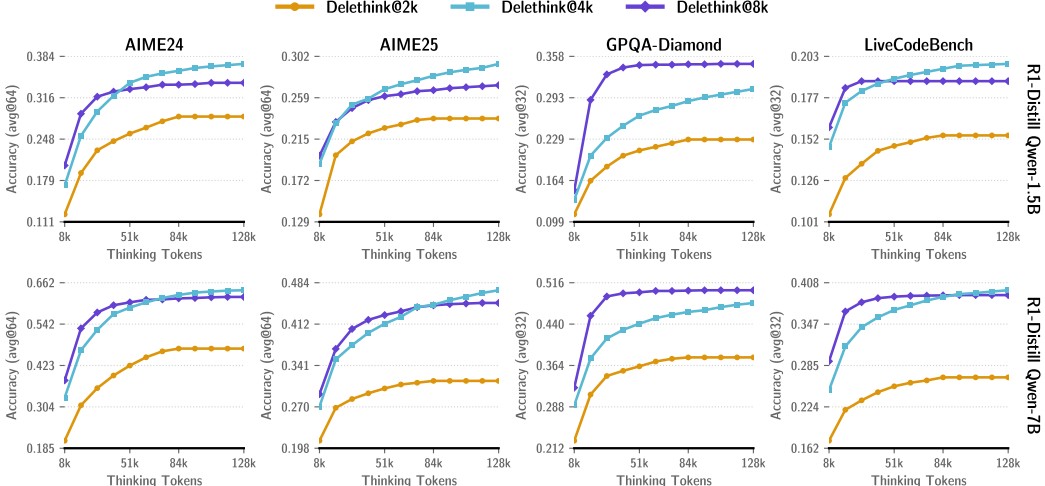

Figure 3: Delethink Inference on R1Distill Models: We ablate among three variants of Delethink where we change the context length. That is, for Delethink@8k we keep 4k tokens and generate 4k tokens, etc. All variants improve performance with more thinking. However, the 4k, test-time scales the performance above the original model's.

et al., 2024; Liu et al., 2024b). These techniques are orthogonal to Delethink: they reduce the cost of a long chain-of-thought by selecting or compressing the KV cache at inference time, whereas we restructure and learn a reasoning process so only a short context is needed in the first place. Recent approaches introduce additional gates to learn how to evict tokens during training (Łańcucki et al., 2025) but usually require introducing additional learnable weights to modify the model architecture. Our approach can be readily applicable to any generation model.

## 3 DELETHINK INFERENCE

In standard LongCoT, sampling a trace $\boldsymbol{y}$ for a given prompt $\boldsymbol{x}$ (i.e., $\boldsymbol{y} \sim \pi(\cdot|\boldsymbol{x})$) the model's context grows linearly with the number of thinking tokens: generating the last token uses $|\boldsymbol{q}|+|\boldsymbol{y}|-1$ tokens. In contrast, Delethink reasons in a *sequence of chunks* with a fixed per-chunk thinking context size $\mathcal{C}$ (e.g., 8K).

Let $\boldsymbol{q}$ denote the query, typically containing a system prompt and a question. In the first chunk, the model generates a response up to $\mathcal{C}$ tokens as usual with the input query as prompt: $\boldsymbol{y}_1 \sim \pi(\cdot|\boldsymbol{x}_1 = \boldsymbol{q})$, where $\boldsymbol{x}_1$ and $\boldsymbol{y}_1$ represent the first chunk's prompt and response, respectively. If the response $\boldsymbol{y}_1$ ends with [EOS], the trace is complete. Otherwise, Delethink advances by constructing the next prompt from the original query and a *markovian state* consisting of the last $m < \mathcal{C}$ tokens of the previous chunk output[2]:

$$\boldsymbol{x}_l = \boldsymbol{q} \oplus \boldsymbol{y}_{(l-1)}[-m :], \quad l \geq 2,$$

where $\oplus$ denotes concatenation and $\boldsymbol{x}_l$ and $\boldsymbol{y}_l$ is the prompt and response for chunk $l$, respectively. In effect, preceding reasoning tokens are deleted when forming the next prompt (hence the name Delethink). Given $\boldsymbol{x}_l$, the model generates up to $\mathcal{C} - m$ new thinking tokens for chunk $l$: $\boldsymbol{y}_l \sim \pi(\cdot|\boldsymbol{x}_l)$. This procedure repeats until [EOS] is produced or the iteration cap $\mathcal{I}$ is reached. We refer to the resulting sequence as a *Delethink trace* $\tau = [(\boldsymbol{x}_1, \boldsymbol{y}_1), \ldots, (\boldsymbol{x}_L, \boldsymbol{y}_L)]$, where $L$ is number of chunks in the Delethink trace $\tau$. We illustrate this process in Figure 1.

In total, the LLM may think up to $\mathcal{C} + (\mathcal{I} - 1)(\mathcal{C} - m)$ tokens, while each chunk's context remains bounded by $|\boldsymbol{q}| + \mathcal{C}$. Since query remains constant through the Delethink process and $|\boldsymbol{q}| \ll \mathcal{C}$ in practice, the maximum per-chunk model context is $O(\mathcal{C}) = O(1)$ with respect to the total number of thinking tokens. In the paper, we fix $m = \mathcal{C}/2$; Appendix B.5 ablates the markovian state size.

---

[2]In practice, we fold the first hundred tokens of the initial chunk into $\boldsymbol{q}$ as it may contain planning tokens.

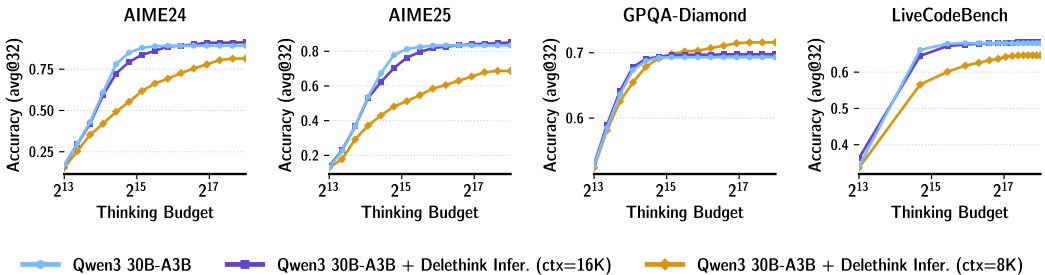

Figure 4: Delethink Inference on Qwen3. Delethink with 16k context length matches the performance of original model that has an 256k context length. Delethink with 8k still scales but does not show the same scaling profile. The reasoning of Qwen3 requires 16k context to become markovian.

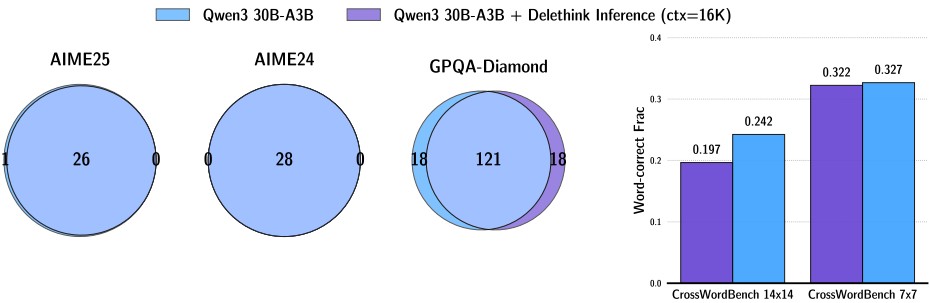

Figure 5: (**Left**) Problem-solving overlap of Delethink vs. LongCoT: on AIME '24 and '25 they solve nearly the same set of questions. On GPQA, each solves an equal number that the other misses. (**Right**) CrossWordBench stress-tests Delethink: deleting previous tokens removes access to already found words. Delethink remains competitive, but its zero-shot limits are evident.

## 3.1 ZERO-SHOT MARKOVIAN THINKING FOR OFF-THE-SHELF LLMS

We test whether Delethink Inference can induce *markovian* thinking *without* additional training or prompting, by applying it to off-the-shelf reasoning LLMs and comparing against standard Long-CoT. We evaluate Qwen3-30B-A3B Thinking (Yang et al., 2025)[3] and the R1-Distill family (Guo et al., 2025) (1.5B–14B), whose native context windows are 256K and 128K, respectively. We evaluate tasks that require extended reasoning: AIME'24 and AIME'25 (MAA, 2025; competition mathematics), GPQA-Diamond (Rein et al., 2024; PhD-level questions), LiveCodeBench[4] (Jain et al., 2025; competition-level programming puzzles). For a controlled comparison, we vary the thinking budget from 8K to 256K tokens (up to 128K for R1-Distill). For Delethink, we additionally vary the per-chunk context $\mathcal{C}$ from 8K to 16K (4K to 8K for R1-Distill since it has a smaller context window) to measure sensitivity to chunk size.

**Results and Test-time scaling**  Delethink Inference matches or exceeds LongCoT across models and tasks (Figures 3 and 17), despite deviating from the training-time regime of the base model. With $\mathcal{C} = 8$k, R1-Distill (1.5B-14B) attains performance and scaling comparable to (and sometimes better than) LongCoT. Qwen3-30B-A3B shows the same pattern with $\mathcal{C} = 16$k. These results indicate that current reasoning LLMs exhibit latent markovian behavior even without explicit training for it. On R1-Distill, Delethink uses additional thinking tokens more effectively: accuracy continues to improve up to 128K tokens, solving instances that LongCoT fails to solve. In contrast, LongCoT plateaus around $\sim 40$K tokens across benchmarks.[5] For Qwen3-30B-A3B the improvement with larger budgets is present but less pronounced.

---

[3]https://huggingface.co/Qwen/Qwen3-30B-A3B-Thinking-2507
[4]2024-08-01 to 2025-02-01, following Guo et al. (2025).
[5]We omit budget-forcing test-time scaling methods such as S1 (Guo et al., 2025) because they hurt performance on reasoning LLMs; see Appendix B.6.2.

---

**Algorithm 1** Delethink Training Step

---

**Inputs:** query $\boldsymbol{q}$; reasoning LLM $\pi_\theta$; thinking context size $\mathcal{C}$; markovian state size $m$; Delethink iterations cap $\mathcal{I}$; group size $G$; reward function $\mathcal{R}$; learning rate $\eta$.

$T, R \leftarrow [\,], [\,]$       ▷ Delethink traces, rewards

**Generate Delethink Traces:**

**for** $i \leftarrow 1$ to $G$ **do**

    $\boldsymbol{x} \leftarrow \boldsymbol{q}$       ▷ prompt

    $\boldsymbol{y} \leftarrow \pi_\theta(\boldsymbol{x}; \mathcal{C})$       ▷ generate up to $\mathcal{C}$ tokens

    $\boldsymbol{q} \leftarrow \boldsymbol{q} \oplus \boldsymbol{y}_{1:100}$       ▷ concatenate first few tokens of $\boldsymbol{y}$

    $\tau \leftarrow [(\boldsymbol{x}, \boldsymbol{y})]$       ▷ trace for group $i$

    **for** $t \leftarrow 1$ to $\mathcal{I} - 1$ **do**

        **if** $\text{last}(\boldsymbol{y}) = [\text{EOS}]$ **then break**

        $\boldsymbol{x} \leftarrow \boldsymbol{q} \oplus \boldsymbol{y}_{-m:}$       ▷ keep last $m$ thinking tokens

        $\boldsymbol{y} \leftarrow \pi_\theta(\boldsymbol{x}, \mathcal{C} - m)$       ▷ generate up to $\mathcal{C} - m$ tokens

        $\text{append}(\tau, (\boldsymbol{x}, \boldsymbol{y}))$       ▷ appending chunk to trace

    $\text{append}(R, \mathcal{R}(\tau))$       ▷ trace reward

    $\text{append}(T, \tau)$

**Estimate Advantages:**

$\{\hat{A}[i]\}_{i=1}^{G} \leftarrow \text{ComputeAdvantage}\left(\{R[i]\}_{i=1}^{G}\right)$       ▷ off-the-shelf advantage estimator

**Updating Parameters:**

$J \leftarrow \frac{1}{G} \sum_{\tau_g} \frac{1}{\ell(\tau_g)} \sum_{l=1}^{|\tau_g|} \mathcal{U}(\boldsymbol{x}_l, \boldsymbol{y}_l; \theta)$       ▷ Compute the loss according to Equation (1)

$\theta \leftarrow \theta + \eta \nabla_\theta J$

---

**Effect of per-chunk context size $\mathcal{C}$**   Smaller $\mathcal{C}$ (e.g., 4K for R1-Distill, 8K for Qwen3-30B-A3B) underperforms larger $\mathcal{C}$ and the LongCoT baseline, as the markovian state $m$ carried between chunks becomes too small and the model can lose the thread of reasoning. Interestingly, smaller $\mathcal{C}$ (e.g., 4K for R1-Distill, 8K for Qwen3-30B-A3B) often yields *better scaling*: because the model is unaware of the inference procedure, it appears to perceive more headroom, emits [EOS] less frequently, and thus uses the full thinking budget more consistently.

**Problem Coverage**   We compare problem coverage on AIME'24, AIME'25, and GPQA-Diamond (Fig 5), counting a problem as solved if the majority vote is correct.[6] On both AIME's, both methods solve essentially the same problems (except 1 on AIME'25). On GPQA-Diamond, 77% of problems are solved by both, but each method uniquely solves about 11% that the other misses.

**Limits of zero-shot Delethink**   To probe failure modes, we evaluate Qwen3-30B-A3B on Cross-WordBench (Leng et al., 2025), which contains crossword puzzles at varying difficulty levels. Reasoning in this tasks requires maintaining a live grid plus filled entries—state that can exceed the capacity of $m$. As shown in Figure 5, Delethink matches LongCoT on $7{\times}7$ puzzles but lags on $14{\times}14$, especially with $\mathcal{C} = 8k$. This suggests that fully native markovian thinkers are needed when the task-state cannot be compressed into the carried context, and that zero-shot Delethink reaches its limits in such settings.

## 4   DELETHINK TRAINING

The RL objective for LLMs is to maximize the expected reward:

$$\mathcal{J}(\theta) = \mathbb{E}_{\boldsymbol{q} \sim \mathcal{D}, \tau \sim \pi_\theta(\cdot | \boldsymbol{q})} \left[\mathcal{R}(\tau)\right] - \beta \, \text{KL}[\pi_\theta \| \pi_{\text{ref}}],$$

where $\mathcal{D}$ is a dataset of queries, $\pi_\theta$ is the policy, $\pi_{\text{ref}}$ is the reference policy, $\beta$ is KL coefficient, $\tau$ is a generated trace using Delethink algorithm (Section 3), and $\mathcal{R}(\tau)$ is the trace-level reward. To optimize this objective, we derive the policy gradient under Delethink dynamics in Appendix B.1. The resulting objective closely mirrors standard RL training for LLMs (Lambert et al., 2024).

---

[6]Majority voting does not apply to code, so LiveCodeBench is excluded.

| Metric | Base | LongCoT | Delethink |
|---|---|---|---|
| Think Tokens | $n$ | $nS$ | $nS$ |
| FLOP | $O(n^2)$ | $O(n^2S^2)$ | $O(n^2S)$ |
| Memory | $O(n)$ | $O(nS)$ | $O(n)$ |
| Backward Time | $T_B$ | $O(T_BS^2)$ | $O(T_BS)$ |
| Generation Time | $T_G$ | $O(T_GS^2)$ | $O(T_GS)$ |

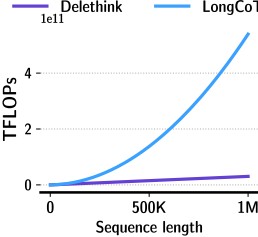

Figure 6: Computational profiles of LongCoT and Delethink scaling from $n$ to $nS$ tokens.

Concretely, Delethink Training iterates three steps: (1) sample Delethink traces for queries, (2) estimate advantages for those traces, and (3) update the model to upweight advantageous tokens and downweight non-advantageous ones. Intuitively, If a trace yields a correct answer, a gradient step increases the probability of all chunks in that trace, since each contributes to the reward.

Given a query $\boldsymbol{q}$, we sample a group $G$ of Delethink traces $\tau_1, \ldots, \tau_G$ from the current policy $\pi_{\theta_{\text{old}}}$, where each trace is a sequence of chunks $\tau = \{(\boldsymbol{x}_l, \boldsymbol{y}_l)\}_{l=1}^{|\tau|}$. Optimizing the expected return proceeds by taking gradients with respect to the following loss function:

$$J(\theta) = \mathbb{E}_{\tau_1,\ldots,\tau_G \sim \pi_{\theta_{\text{old}}}(.|\boldsymbol{q})}\left[\frac{1}{G}\sum_{\tau_g}\underbrace{\frac{1}{\ell(\tau_g)}\sum_{l=1}^{|\tau_g|}\mathcal{U}(\boldsymbol{x}_l, \boldsymbol{y}_l; \theta)}_{\text{per-Delethink trace loss}}\right]. \tag{1}$$

where $\ell(\tau) = \sum_l |\boldsymbol{y}_l|$ is the total number of response tokens in a Delethink trace $\tau$. This is similar to GRPO, where per-trace term is normalized by the total number of thinking tokens. $\mathcal{U}(\boldsymbol{x}, \boldsymbol{y}; \theta)$ represents the per-chunk $(\boldsymbol{x}, \boldsymbol{y})$ objective which closely follow that of PPO in LLMs. Specifically,

$$\mathcal{U}(\boldsymbol{x}, \boldsymbol{y}, \theta) = \sum_{t=1}^{|\boldsymbol{y}|}\min\left[\frac{\pi_\theta(y_t)}{\pi_{\theta_{\text{old}}}(y_t)}\hat{A}_t, \text{clip}\left(\frac{\pi_\theta(y_t)}{\pi_{\theta_{\text{old}}}(y_t)}, 1-\epsilon, 1+\epsilon\right)\hat{A}_t\right] - \beta\text{KL}[\pi||\pi_{\text{ref}}],$$

where $\pi(y_t)$ is the probability of predicting token $y_t$ of the chunk's response $\boldsymbol{y}$,[7] KL is Kullback–Leibler loss (Shao et al., 2024) controlled by coeffcient $\beta$. The advantage $\hat{A}_t$ can be estimated with any off-the-shelf estimator (Kazemnejad et al., 2025; Yu et al., 2025). For simplicity we use the GRPO formulation: $A_{l,t} \equiv A_{\tau_g} := (\mathcal{R}(\tau_g) - \mu)/\sigma$, where $\mathcal{R}(\tau_g)$ is the reward for $g$-th trace (e.g. whether model reaches the correct answer at the end of last chunk). $\mu$ and $\sigma$ are the mean and standard deviation of rewards across the trace group $\{\tau_g\}$. Pseudo-code is shown in Algorithm 1, illustrating training on a single query (we optimize over batches in practice).

### 4.1 COMPUTATIONAL COST OF SCALING THINKING

An RL step has two parts: generation and backward for updating the policy. We study how both scale when an LLM's thinking length grows from $n$ tokens to $nS$ tokens under LongCoT and Delethink. As Table 6 shows, both stages scale linearly in Delethink but quadratically in LongCoT.

**Total FLOPs and Backward Time** Suppose training an LLM to think for $n$ tokens costs $C$ FLOPs. With LongCoT, scaling by $S$ costs $O(CS^2)$ because self-attention grows quadratically with length. Delethink instead runs in $2S$ chunks. Each chunk carries forward $\frac{n}{2}$ tokens and generates $\frac{n}{2}$ new ones[8]. The result is $O(4n^2S)$, linear in $S$. Backward time tracks total FLOPs.

**Peak memory** KV Cache entries have fixed size per token. In LongCoT, the KV cache grows linearly with thinking length, limiting parallel requests on the GPU. For example, the KV cache of

---

[7]We omit conditioning on the context $\boldsymbol{x}, \boldsymbol{y}_{<t}$ for brevity.
[8]Assume $m = \mathcal{C}/2$

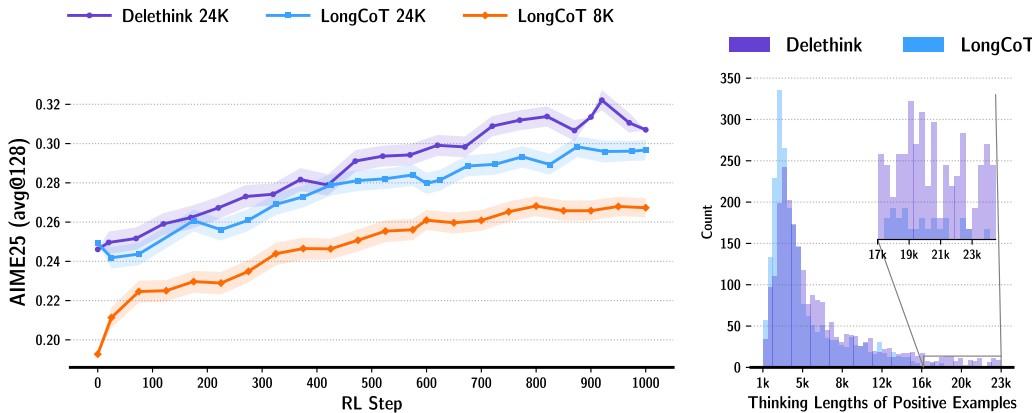

Figure 7: (**Left**) Delethink Training and LongCoT both train R1DistillQwen1.5B to think for 24k tokens. Delethink learns as effectively as LongCoT even though it uses 70% less memory and generates tokens 40% faster. (**Right**) Delethink and LongCoT use their thinking budgets well. At longer lengths, Delethink produces more correct answers, showing it spends its budget effectively.

a 1M-token trace on R1-Distill 1.5B, a small model, alone fills an entire H100. Going beyond requires sharding the sequence across GPUs with sequence-parallelism, adding heavy communication overhead. In contrast, Delethink keeps only the current chunk's KV cache, so usage stays constant.

**Generation Time** Assume an infinite stream of requests saturating the GPU, each with maximum length $L$. The optimal throughput satisfies $T \propto \frac{1}{L}$ under simplifying assumptions (Ao et al., 2025; see B.2). This can be intuited from the fact that generation must access the KV cache, which grows linearly with sequence length, limiting parallel requests. LongCoT uses a growing KV cache; Delethink keeps it fixed per chunk. Both must generate $nS$ tokens, but LongCoT's throughput falls by a factor of $S$, scaling time by $S^2$. Delethink's throughput is constant, so its time scales only as $S$.

## 5  RL TRAINING EXPERIMENTAL SETUP

**Model and Dataset.** We train R1DistillQwen1.5B (Guo et al., 2025) on the DeepScaleR dataset (Luo et al., 2025b), which contains 40k verifiable math questions. We benchmark our models on AIME'24, AIME'25 (MAA, 2025), and HMMT.

**Baselines.** Our main baseline trains with a maximum of 24k thinking tokens. We also train a baseline with 8k thinking tokens to show the benefit of allowing more thinking tokens. We refer to the first as LongCoT-24k and the latter as LongCoT-8k

**Training Setup.** We set Delethink cap iteration, $\mathcal{I} = 5$, and a thinking context size,$\mathcal{C} = 8$k tokens. With this setup, the model is able to think up to 24k tokens which equals LongCoT's thinking budget. We train all models for 1000 RL steps with a learning rate of $1\mathrm{e}{-}6$. In each step we sample 128 questions and generate 8 responses per question. Delethink is implemented in the verl framework (Sheng et al., 2024) and uses sglang (Zheng et al., 2023) for generating responses. Training runs on 8 H100 GPUs without sequence parallelism. See B.4 for full details.

## 6  DELETHINK TRAINING RESULTS

Delethink and LongCoT-24k share a 24k-token thinking budget and start at the same pre-training performance (Figure 7), showing that Delethink turns R1DistillQwen1.5B into a Markovian thinker on par with its LongCoT counterpart (Section 3). Despite each Delethink RL step costing less than a LongCoT-24k step, Delethink matches and at equal RL steps, surpasses LongCoT-24k during training (Figure 7). Thus, Delethink learns as effectively as LongCoT while using fewer compute resources. The extended budget size is critical. With only 8k tokens, LongCoT underperforms Delethink by 5.5% (Figure 7), confirming that the additional 16k thinking tokens are necessary for

the stronger results and that Delethink learns to leverage them. Figure 8 supports this as it shows that both LongCoT-24k and Delethink increase their accuracy while spending their thinking budget.

Training also induces budget awareness. As discussed in Section 3, zero-shot application of Delethink Inference might over-iterate or under-iterate when the maximum iteration is unknown. Our RL scheme penalizes traces that exceed the maximum iterations and rewards those that respect it. Over training, successful responses lengthen as the thinker exploits its extra thinking budget (Figure 7), while negatively rewarded, overthought traces shorten.

### 6.1 EMPIRICAL COMPUTE

We evaluate the compute efficiency of Delethink against LongCoT-24k. Delethink completes each RL step in 215s, faster than LongCoT-24k at 248.5s. Its token generation is also more efficient, reaching 8,500 tokens per second per H100 versus 6,000 for LongCoT-24k. As a result, Delethink finishes batch response generation in 130s compared to LongCoT's 170s, in line with the predictions of Section 4.1. The backward pass takes 80s for Delethink while 70s for LongCoT-24k. At first glance, this appears to contradict theory. However, the difference arises from constant and lower-order terms that dominate at shorter sequence lengths. Figure 11 shows the crossover for R1DistillQwen1.5B. below 32k tokens, Delethink can be slower since non-attention blocks dominate, but beyond that point, the quadratic scaling of attention makes Delethink increasingly faster. At one million tokens, Delethink achieves $17\times$ reduction in FLOPs. In Figure 2, we empirically measure the time of a single RL step assuming a target average thinking length. As theory predicts, time grows quadratically for LongCoT but linearly for Delethink. Additionally, We also measure the throughputs of inference engines under both methods. Consistent with the theory, Delethink maintains the same throughput regardless of thinking length, while LongCoT shows a steady decline.

### 6.2 DELETHINK TEST-TIME SCALING

**Delethink Inference vs. LongCoT** We investigate the test-time scaling behavior of Delethink and LongCoT. As shown in Figure 2, both LongCoT-24k and LongCoT-8k quickly plateau once they reach their training-time limits, indicating that LongCoT's test-time scaling is largely constrained by its training budget. In contrast, Delethink continues to improve even when reasoning with 100k more tokens than it encountered during training. For example, some AIME'25 problems are only solved after 140k tokens B.7, despite the model being trained on just 24k. This demonstrates that Delethink enables genuine test-time scaling of thinking tokens far beyond its training-time limits.

**Delethink Inference on LongCoT Checkpoints** As discussed in Section 3.1, off-the-shelf LLMs behave as Markovian thinkers under Delethink. This lets us generate traces from models not trained with it. That suggests Delethink Inference may improve the LongCoT-24k checkpoint though it was not trained with Delethink. We test this in Figure 15. Delethink Inference lets the LongCoT-24k checkpoint scale past its training limit at test time. Performance rises by nearly $4\%$, almost matching the gain from RL training, with no extra training cost. We hypothesize that keeping queries under 8k tokens makes the 24k baseline behave as if it has room to continue. But when queries stay under 16k, the effect disappears, breaking the *infinite budget illusion*. Delethink Inference might improve conventionally trained checkpoints, though they cost quadratically more to train than Delethink.

### 6.3 CONTEXT SIZE ABLATION

We ablate context size to 4K and 2K, set the state size $m$ to half the context size, and adjust the number of iterations to keep the maximum thinking length at 24K tokens. We call these runs 8K, 4K, and 2K. As Figure 16 shows they successfully train with smaller state for Markovian Thinking.

### 6.4 SCALING DELETHINK TO 96K

We scale the thinking budget from 24K to 96K tokens to test Delethink's scalability. We fix the context at $\mathcal{C} = 8$K and raise the iteration cap from $\mathcal{I} = 5$ to $\mathcal{I} = 23$, yielding a 96K total budget. We train on OpenMath (Moshkov et al., 2025), a harder benchmark than DeepScaleR that demands longer reasoning. Starting from the 24K checkpoint, we continue training for 150 steps with unchanged hyperparameters. Figure 10 reports results on AIME'24 and AIME'25. Delethink 96K

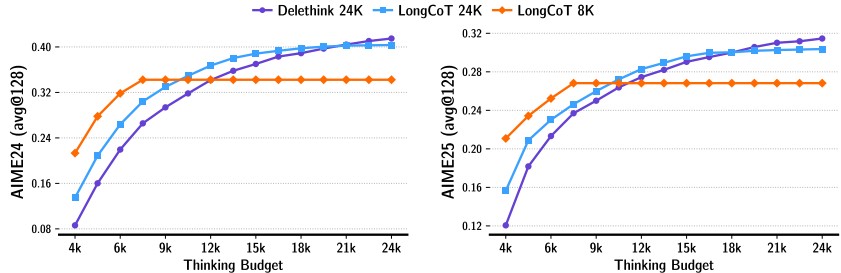

Figure 8: Cumulative accuracy on AIME'24: LongCoT-8k plateaus once it exceeds its 8k training-time limit. In contrast, Delethink and LongCoT-24k learn to use their thinking budget effectively.

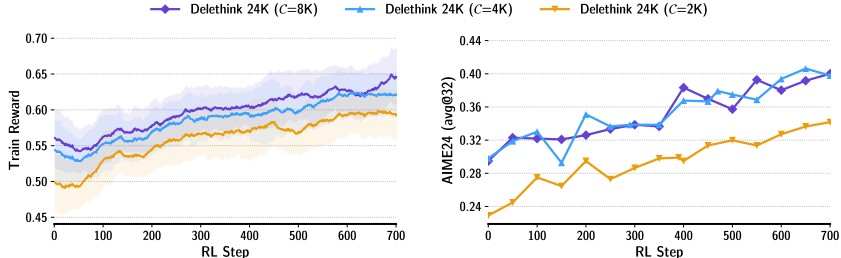

Figure 9: We vary $\mathcal{C}$ while fixing the thinking budget at 24K tokens. **(Left)** Smoothed training reward vs. RL step. **(Right)** Validation accuracy (AIME'24) vs. RL step. Delethink 24K with $\mathcal{C} = 8$K and 4K performs similarly. The 2K variant lags but improves steadily over the base model via training.

surpasses the 24K checkpoint and matches or exceeds 24K with test-time scaling to 128K tokens. The average reasoning length reaches 36K tokens on AIME'24 and 42K on AIME'25, indicating active use of the larger budget. These results show that Delethink scales to long reasoning traces.

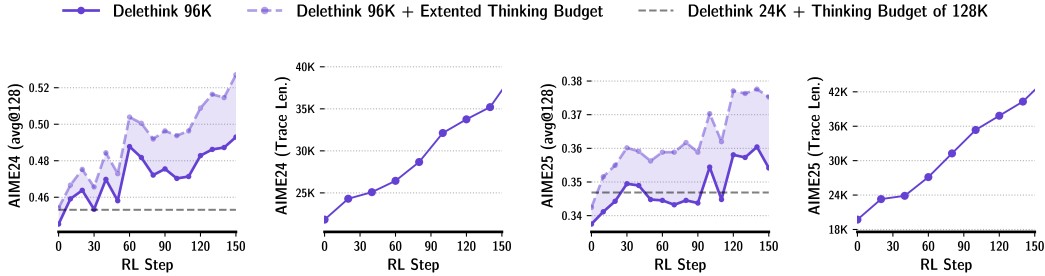

Figure 10: **Scaling Delethink to 96K.** AIME'24/'25 accuracy and average trace length vs. RL step for Delethink 96K; dashed curves show extended thinking budget. Despite only 150 RL steps, 96K surpasses both baseline and its extended thinking variant, with mean trace lengths up to 42K.

# 7 DISCUSSION

Extending thinking length has advanced reasoning in LLMs, but at quadratic compute cost. We propose the Markovian Thinking Paradigm, where each step retains only the minimal state needed to continue, yielding linear compute and constant memory for both training and inference. We show it is practical: off-the-shelf LLMs become Markovian via Delethink Inference and can be further refined into native Markovian thinkers with RL through Delethink Training. Experiments match LongCoT performance while using linear compute and fixed memory, and Delethink Inference enables test-time scaling of thinking length beyond static LongCoT budgets. Thus, Markovian Thinking moves from concept to practice, opening a path to reasoning at tens of millions of tokens.

ACKNOWLEDGEMENTS

SR is supported by a CIFAR AI Chair and a Mila–Samsung grant. AC is supported by Canada CIFAR AI Chair, the Canada Research Chair, the NSERC Discovery Grant, and funding from Microsoft Research. SC is supported by the Canada CIFAR AI Chair, the Canada Research Chair in Lifelong Machine Learning, and the NSERC Discovery Grant. We thank the Mila IDT team and the Digital Research Alliance of Canada for providing the compute resources used in our experiments.

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

## A  APPENDIX

## B  LLM USAGE

We used ChatGPT to refine the English writing of the paper.

### B.1  DERIVING DELETHINK LOSS

In this section, we show that the Delethink Training loss is basically the policy gradient. Note that the Policy gradient without the PPO clipping is the following:

$$\nabla_\theta J(\theta) = \mathbb{E}_{\tau \sim \pi_\theta} \left[ \sum_{t=0}^{T} \nabla_\theta \log \pi_\theta(a_t \mid s_t) A_t \right]. \tag{2}$$

Here, $\tau = (s_0, a_0, r_0, \ldots, s_T)$ denotes a trajectory sampled from the policy, and $A_t$ is the advantage at time $t$. In Delethink, the actions are generating the response segment of each chunk given the

query segment. Note that we need to sum this across the Delethink trace. Plugging in the policy gradient would be,

$$\nabla_\theta J(\theta) = \mathbb{E}_{\tau \sim \pi_\theta} \sum_{i=1}^{|\tau|} \sum_{t=1}^{|\mathbf{y}^{(i)}|} \hat{A}_t \ \log \pi_\theta\big(y_t \mid \mathbf{x}^{(i)}, \mathbf{y}_{<t}^{(i)}\big) \tag{3}$$

. If we add the PPO Clipping and the ratios to this objective it results in:

$$\mathcal{U}(\tau, \theta) = \sum_{i=1}^{|\tau|} \sum_{t=1}^{|\boldsymbol{y}^{(i)}|} \min \left[ r_\theta^t(\boldsymbol{y}^{(i)}, \boldsymbol{x}^{(i)}) \hat{A}_t, \operatorname{clip}\left( r_\theta^t(\boldsymbol{y}^{(i)}, \boldsymbol{x}^{(i)}), 1 - \epsilon, 1 + \epsilon \right) \hat{A}_t \right] \tag{4}$$

where

$$r_\theta^t(\boldsymbol{y}^{(i)}, \boldsymbol{x}^{(i)}) = \frac{\pi_\theta\left( y_t^{(i)} \mid \boldsymbol{x}^{(i)}, \boldsymbol{y}_{<t}^{(i)} \right)}{\pi_{\theta_{\text{old}}}\left( y_t^{(i)} \mid \boldsymbol{x}^{(i)}, \boldsymbol{y}_{<t}^{(i)} \right)}. \tag{5}$$

The only extra term that is left is the KL term which is added only to constrain the policy not to deviate from the base model. With that addition, we will have:

$$\nabla_\theta J_{\text{KL}}(\theta) = \mathbb{E}_{\tau \sim \pi_\theta} \sum_{i=1}^{|\tau|} \sum_{t=1}^{|\boldsymbol{y}^{(i)}|} \min \left[ r_\theta^t(\boldsymbol{y}^{(i)}, \boldsymbol{x}^{(i)}) \hat{A}_t, \operatorname{clip}\left( r_\theta^t(\boldsymbol{y}^{(i)}, \boldsymbol{x}^{(i)}), 1 - \epsilon, 1 + \epsilon \right) \hat{A}_t \right] - \beta KL[\pi || \pi_{\text{ref}}] \tag{6}$$

That concludes our proof.

## B.2 Deriving Throughput Relation

We follow Ao et al. (2025). Consider a GPU with maximum memory $M$, serving an infinite stream of incoming requests and an equally unbounded stream of completed outputs. Let $n^*$ denote the equilibrium number of concurrent requests. Each request has $l$ prefill tokens and $l'$ decode tokens. In steady state, the KV-cache memory required is

$$M^* = n^* \left( l + \frac{l'}{2} \right). \tag{7}$$

According to Ao et al. (2025), the equilibrium throughput of an attention-based LLM under a memory constraint is

$$\mathcal{T}^* = \frac{n^*}{d_0 + d_1 n^* \left( l + \frac{l'}{2} \right)}, \tag{8}$$

where $d_0$ is the fixed per-batch overhead and $d_1$ is the time cost per unit of memory. Thus, throughput exhibits an inverse dependence on the total effective context length. In regimes where $d_1 n^* \left( l + \frac{l'}{2} \right) \gg d_0$, we obtain the approximation

$$\text{Throughput}^* \approx \frac{n^*}{d_1 n^* \left( l + \frac{l'}{2} \right)} = \frac{1}{d_1 \left( l + \frac{l'}{2} \right)}. \tag{9}$$

Consequently, for the same GPU, the ratio of throughputs when decoding lengths are $l_1'$ and $l_2'$ satisfies

$$\frac{\mathcal{T}_{l_1'}}{\mathcal{T}_{l_2'}} \approx \frac{\frac{1}{d_1 \left( l + \frac{l_1'}{2} \right)}}{\frac{1}{d_1 \left( l + \frac{l_2'}{2} \right)}} = \frac{l + \frac{l_2'}{2}}{l + \frac{l_1'}{2}}.$$

In the long-thinking regime where $l' \gg l$, this simplifies to

$$\frac{\mathcal{T}_{l_1'}}{\mathcal{T}_{l_2'}} \approx \frac{l_2'}{l_1'},$$

establishing that throughput is (approximately) inversely proportional to the decoding length in this regime. □

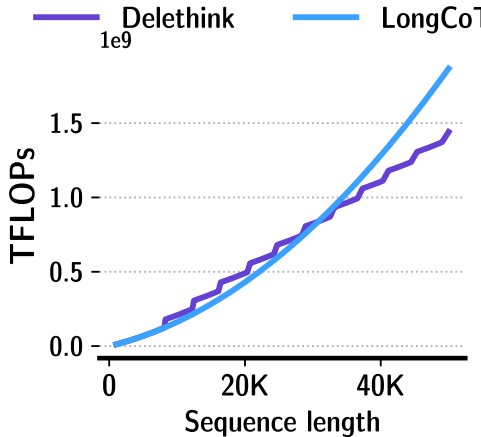

Figure 11: When training R1DistillQwen1.5B with Delethink with 8k thinking context, for thinking lengths $\sim$ 30k the number of FLOPs for both is equal. This is because the non-attention components like dense layers dominate here. However, after 30k quadratic cost of attention dominates.

**Intuition.** Decoding repeatedly accesses the KV cache, whose size, and therefore memory-time cost, grows linearly with the number of decoded tokens $l'$. This linear growth drives the inverse relationship between throughput and $l'$ in the long-thinking regime.

### B.3    DELETHINK FLOPS CROSSOVER VS. LONGCOT24K

We can compute, in closed form, the FLOPs required for reasoning of a certain length under Delethink and LongCoT. We calculated this for our main experiment setup, where Delethink runs with $\mathcal{C} = 8\text{K}, m = 4\text{K}$, single step with 1000 episodes, and results are shown in Figure 6. We highlight the crossover in Figure 11 at short sequences. Up to roughly 30K tokens, Delethink spends slightly more FLOPs than LongCoT. That seems counterintuitive: Delethink scales linearly, LongCoT quadratically. This is because in Delethink each token is generated once and then reprocessed as input to the next chunk. While the quadratic compute of attention quickly overtakes this double-counting cost in longer sequences, it matters at short lengths. This is clear when considering non-attention layers. FLOPs in non-attention layers scale linearly, and in Delethink each token is generated once and then reprocessed in the next chunk as a prompt. Therefore, the cost of non-attention layers doubles. The crossover occurs around 30K tokens in our setting. Beyond this point, Delethink is linearly more efficient in terms of FLOPs. Note that while total backward time tracks total FLOPs, generation time at these response lengths is still slower for LongCoT. This is because generation throughput is limited by memory-access and memory-footprint bottlenecks, not just FLOPs. As explained in Section 6.1, our training still runs faster even in this regime due to faster generation. Our next step, Delethink 96K, is significantly faster than its counterpart LongCoT-RL 96K in both backward pass and generation, to the point that we could not train the latter under our compute resources.

### B.4    DETAILED RL TRAINING SETUP

In this section we describe our training setup and hyperparameters meticulously.

#### B.4.1    HYPERPARAMETERS

Table 1 represents the key hyperparamters used in our RL training experiments.

### B.5    PREFILL ABLATION OF QWEN3

In Figure 12 we showcase what is necessary carry-over size to make the thinking process markovian for different models.

| Hyperparameter | Delethink Training | RL LongCoT |
|---|---|---|
| **Rollout (inference)** | | |
| Sampling (T / top-p / top-k / $n$) | 0.6 / 1.0 / $-1$ / 8 | 0.6 / 1.0 / $-1$ / 8 |
| Prompt / response length (tokens) | 2,048 / 8,192 | 2,048 / 24,576 |
| **Algorithm** | | |
| Advantage estimator | GRPO | GRPO |
| KL penalty | N/A | N/A |
| **Policy optimization** | | |
| PPO epochs / number of grad updates | 1 / 2 | 1 / — |
| Clip ratio (low / high) | 0.20 / 0.26 | 0.20 / 0.26 |
| Grad clip (global-norm) | 1.0 | 1.0 |
| Loss aggregation | Trace Length-Batch Mean | Seq-mean-Batch Mean |
| **Actor optimizer** | | |
| Learning rate | $1 \times 10^{-6}$ | $1 \times 10^{-6}$ |
| Weight decay | 0 | 0 |
| Warmup | constant; steps $= -1$ (disabled) | constant; steps $= -1$ (disabled) |
| Total training steps (optim) | 1,000 | 1,000 |
| **Data** | | |
| Train / val batch size | 128 / 16 (per step, logical) | 128 / 16 (per step, logical) |
| **Rewarding / critics** | | |
| Reward function | HF-Math-Verify | HF-Math-Verify |

Table 1: Key hyperparameters for **RL LongCoT** vs. **Delethink Training**.

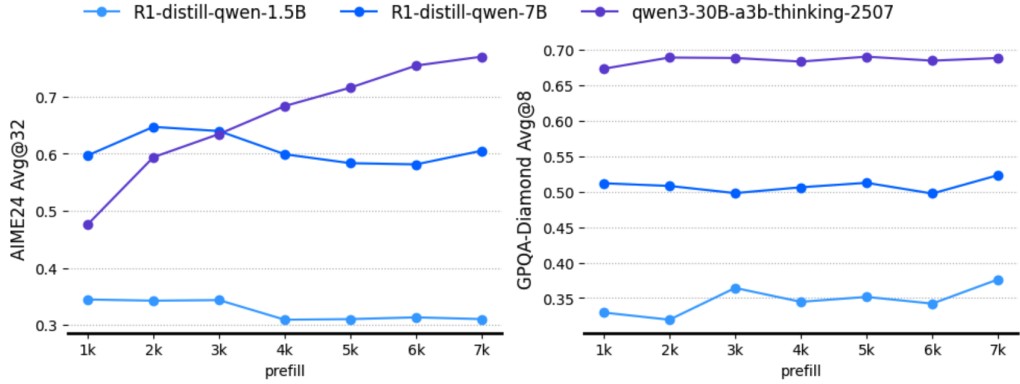

Figure 12: Ablating the necessary state size to make the process markovian. 1B and 7B R1Distill models become markovian even with short context lengthes. Qwen3 which has an original 256k context, requires a larger state size for becoming a markovian thinker.

## B.6 ZERO-SHOT MARKOVIAN THINKING FOR R1DISTILL

### B.6.1 WORKING MEMORY ABLATION

In Figure 13 we ablate the necessary context size and its affect on zero-shot Delethink performance.

### B.6.2 BUDGET FORCE ABLATION: DOES S1 WORK ON R1?

We evaluate *Budget-Force* decoding scheme inspired by the S1 "simple test-time scaling" protocol (Guo et al., 2025). We run decoding and, whenever the model *finalizes early* before using the allotted token budget, we cut the response at the first finalization marker and append a short continuation cue before regenerating from the *entire* prompt. Concretely, for each query we first sample with temperature 0.6, top-$p = 0.95$, $n = 1$ under a 32k token budget. If the model emits any of `</think>`, `**Final Answer**`, or `\boxed{` (or EOS) *before* the budget is consumed, we

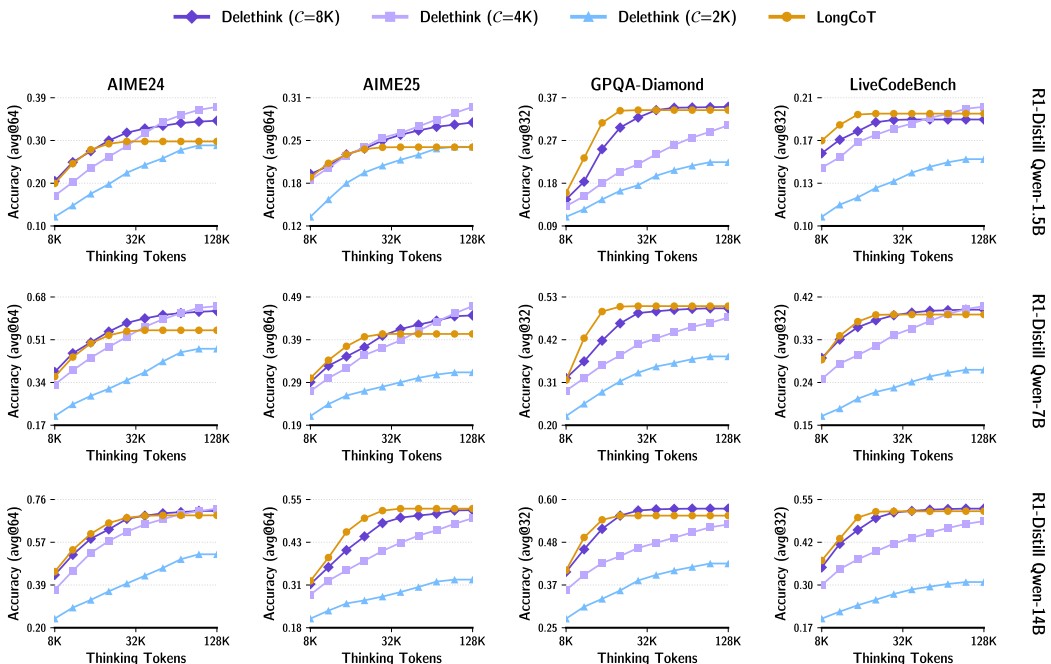

Figure 13: Thinking Context Size Ablation. Scaling behavior of R1-Distill models across AIME'24, AIME'25, GPQA-Diamond, and LiveCodeBench under varying per-hunk contexts $\mathcal{C} \in \{2k, 4k, 8k\}$. Accuracy is plotted against total thinking tokens, with smaller contexts requiring proportionally more iterations.

Table 2: Accuracy (%) across benchmarks under identical compute/decoding budgets.

| Model | Method | AIME24 | AIME25 | GPQA-Diamond | LiveCodeBench |
|---|---|---|---|---|---|
| R1-distill-qwen-1.5B | Long CoT 32k | 28.2 | 24.2 | 33.7 | 18.8 |
| | Budget Force | 24.6 | 19.9 | 33.4 | 17.7 |
| | Delethink | **37.6** | **29.7** | **34.6** | **19.9** |
| R1-distill-qwen-7B | Long CoT 32k | 53.1 | 39.3 | 49.5 | 38.2 |
| | Budget Force | 50.1 | 35.1 | 47.8 | 36.6 |
| | Delethink | **64.8** | **47.5** | **50.1** | **40.1** |
| R1-distill-qwen-14B | Long CoT 32k | 68.9 | **52.9** | **59.1** | 51.71 |
| | Budget Force | 70.2 | 52.6 | 59.1 | 51.9 |
| | Delethink | **72.6** | 52.5 | 57.5 | – |

trim at the earliest occurrence, append the literal cue `Wait\n`, and re-issue a generation from *query + cut_response*. We repeat this micro-forcing until the round budget is exhausted or a continuation cap (20) is reached. To avoid distributional drift, we **do not** alter model decoding settings mid-run; the only intervention is the *prompt-level* cut-and-continue. At the end, if the model still has not emitted EOS after fully using the budget, we add a compact finalization hint (`**Final Answer**\n`) and request a short natural continuation. All models identical sampling parameters across conditions. We compare three settings: (i) **Long-CoT 32k** (single pass, no forcing), (ii) **Budget-Force** as above, and (iii) **Delethink**. Unless noted, we use `continue_iters= 20`, `num_samples= 8`, math tasks are verified with the HF checker using `</think>` as the thinking delimiter. Scores reported in Table 2 are accuracy on AIME'24/AIME'25, GPQA-Diamond, and LiveCodeBench under the same compute budgets. In almost all the cases, Budget Force peforms worse than the LongCoT baseline (normal sampling), demonstrating that such methods are not applicable to reasoning models like R1-distill family.

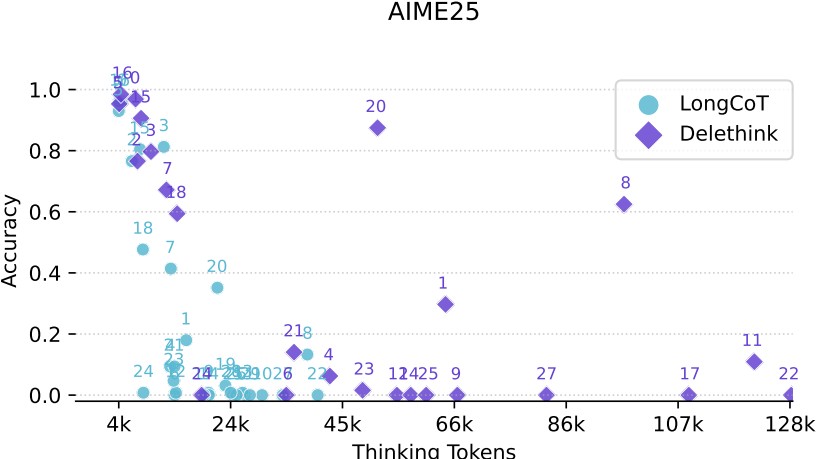

Figure 14: The average thinking length per question and its corresponding accuracy. Delethink Inference truly test-time scales R1DistillQwen1.5B performance on AIME'25.

### B.6.3 HYPERPARAMETERS

### B.7 AIME'25 DETAILED LENGTH VS. ACCURACY

In Figure 14 we show detailed lengths of Delethink traces on solving AIME'25 to showcase that Delethink solves questions via genuine test-time scaling.

## C    DELETHINK INFERENCE ON LONGCOT CHECKPOINTS

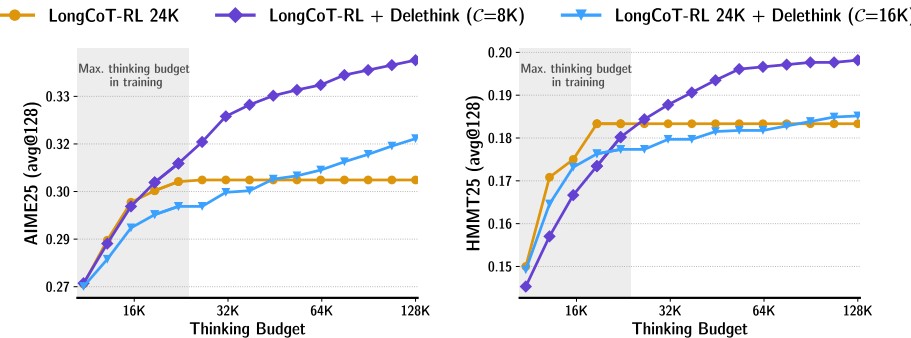

Figure 15: Delethink Inference on a LongCoT–RL 24K checkpoint (AIME'25, avg@128). Long-CoT plateaus near its trained budget, while Delethink ($\mathcal{C} = 8k$, $\mathcal{C} = 16k$) keeps scaling; $\mathcal{C} = 8k$ yields the larger gain. The shaded region marks the trained budget.

## D    CONTEXT SIZE ABLATION RESULTS

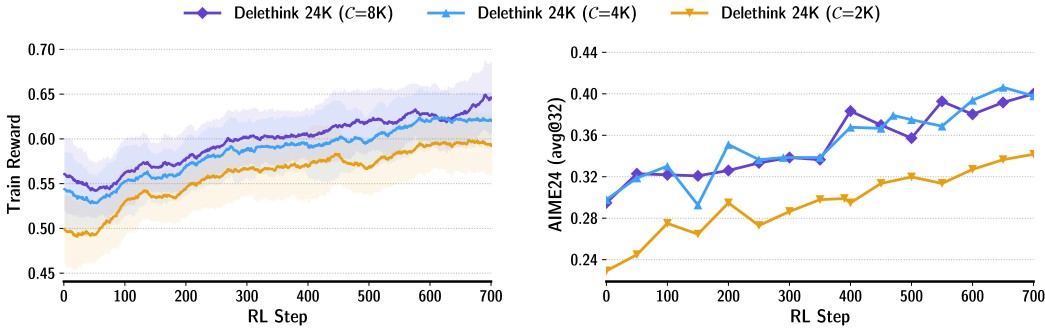

Figure 16: We vary $\mathcal{C}$ while fixing the thinking budget at $\approx 24K$ tokens; smaller $\mathcal{C}$ yields a smaller Markov state. **(Left)** Smoothed training reward vs. RL step. **(Right)** Validation accuracy (AIME'24) vs. RL step. Delethink 24K with $\mathcal{C} = 8K$ and 4K performs similarly in training reward and validation accuracy. The 2K variant lags but improves steadily over the base model during Delethink training.

## E    ZERO-SHOT DELETHINK INFERENCE ON STATE-OF-THE-ART LLMS

In Figure 17 we show zero-shot application of Delethink Inference on off-the-shelf state-of-the-art LLMs.

## F    DETAILED TASK PERFORMANCE

In Figure 18 we show detailed performance of Delethink checkpoints on both in-distribution and out-of-distribution tasks.

## G    DELETHINK EXAMPLES

In Figure 19 and Figure 20 we show Delethink Inference in action.

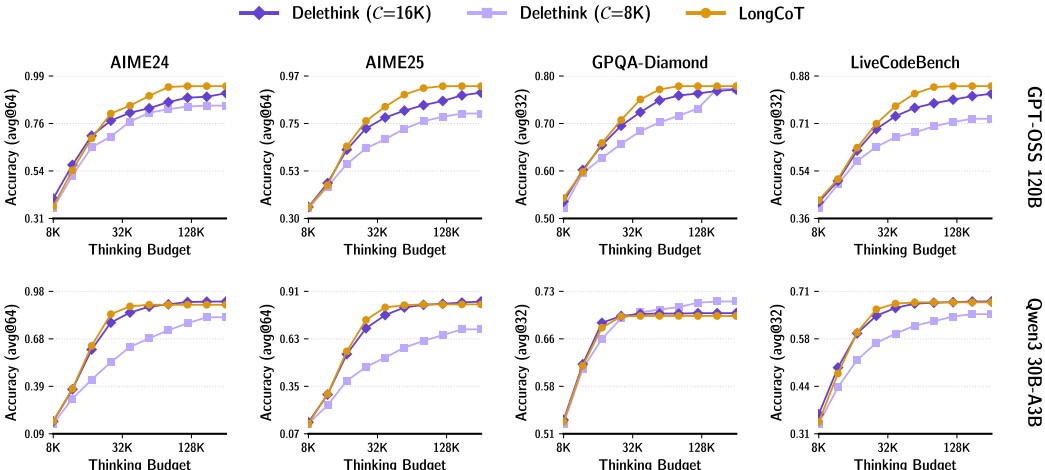

Figure 17: State-of-the-art reasoning LLMs, GPT-OSS-120B and Qwen3-30B-A3B, are capable of Markovian Thinking zero-shot, providing a strong initialization for training, signaling scalability. Delethink closely tracks LongCoT and recovers most of its final accuracy (the curves nearly coincide on Qwen3)

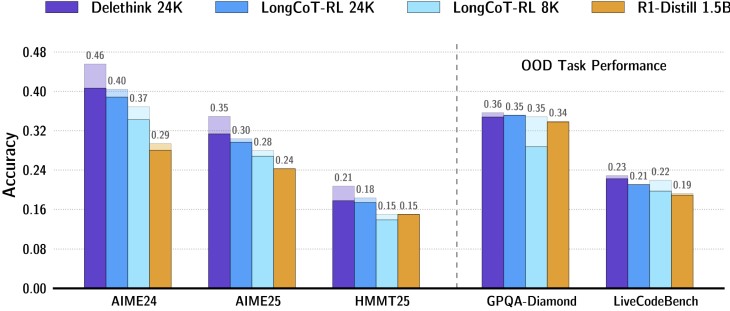

Figure 18: On IID math tasks (AIME'24/'25, HMMT'25), Delethink outperforms LongCoT-RL 24K. Shaded regions show gains from test-time scaling (through sequential sampling), where Delethink improves the performance even more; on OOD tasks (GPQA-Diamond, LiveCodeBench) gains are modest, yet Delethink still matches or slightly beats LongCoT-RL 24K.

**Prompt**

Given that $O$ is any point in space, and $A$, $B$, $C$, $D$ are four points such that no three of them are collinear, but they are coplanar, and $\overrightarrow{OA} = 2x \cdot \overrightarrow{BO} + 3y \cdot \overrightarrow{CO} + 4z \cdot \overrightarrow{DO}$, find the value of $2x + 3y + 4z$.

**Response**

`<think>` Okay, so I've got this vector problem here. Let me read it again and make sure I understand what's being asked.

We have a point O in space, and four points A, B, C, D such that no three of them are collinear, but they are coplanar. That means all four points lie on the same plane, but none of them lie on the same straight line. Interesting. So, they're coplanar, but not colinear.

Then, we're given that the vector OA is equal to 2x times vector BO plus 3y times vector CO plus 4z times vector DO. We need to find the value of 2x + 3y + 4z.

[…]

Alternatively, perhaps if I take another vector product, say the scalar product with $\overrightarrow{A}$, but that might not be helpful.

Wait, perhaps I should consider that the equation must hold for any position of O, which would require that the coefficients multiplying O in the equation must be zero. Wait, let me see:

From:

$$\overrightarrow{A} = (2x + 3y + 4z)\overrightarrow{O} - (2x\overrightarrow{B} + 3y\overrightarrow{C} + 4z\overrightarrow{D})$$

So,

$$\overrightarrow{O} = \frac{\overrightarrow{A} + 2x\overrightarrow{B} + 3y\overrightarrow{C} + 4z\overrightarrow{D}}{2x + 3y + 4z + 1}$$

Since O is arbitrary, the only way this equation holds is if the numerator is proportional to $\overrightarrow{O}$
, but since $\overrightarrow{O}$ is free, perhaps the coefficients must satisfy:
The numerator must be zero, but that seems conflicting.

Wait, maybe not. Alternatively, perhaps if I take two different points, say O itself, and set up equations.

Wait, this is getting a bit tangled. Let's step back.

We have the equation:

$$\overrightarrow{OA} = 2x(\overrightarrow{BO}) + 3y(\overrightarrow{CO}) + 4z(\overrightarrow{DO})$$

[…]

Expressed as vectors:

$$\overrightarrow{A} - \overrightarrow{O} = 2x(\overrightarrow{O} - \overrightarrow{B}) + 3y(\overrightarrow{O} - \overrightarrow{C}) + 4z(\overrightarrow{O} - \overrightarrow{D})$$

Expanding:

$$\overrightarrow{A} - \overrightarrow{O} = 2x\overrightarrow{O} - 2x\overrightarrow{B} + 3y\overrightarrow{O} - 3y\overrightarrow{C} + 4z\overrightarrow{O} - 4z\overrightarrow{D}$$

Bring all $\overrightarrow{O}$ terms to the left:
$\overrightarrow{A} = (2x + 3y + 4z + 1)\overrightarrow{O}$
$-2x\overrightarrow{B} - 3y\overrightarrow{C} - 4z\overrightarrow{D}$

Wait, that seems similar to before. Then, if I rearrange:

$$\overrightarrow{A} + 2x\overrightarrow{B} + 3y\overrightarrow{C} + 4z\overrightarrow{D} = (2x + 3y + 4z + 1)\overrightarrow{O}$$

But since O is arbitrary, the vector on the right can be any vector in space because $\overrightarrow{O}$ can be any vector, but the left side is fixed as $\overrightarrow{A} + 2x\overrightarrow{B} + 3y\overrightarrow{C} + 4z\overrightarrow{D}$. So the only way this can hold
[…] `</think>`

Figure 19: Example of Chunk on boundaries in Delethink-RL. The horizontal line specifies the chunk boundary.

**Prompt**

Suppose that $m$ and $n$ are positive integers with $m < n$ such that the interval $[m, n)$ contains more multiples of 2021 than multiples of 2000. Compute the maximum possible value of $n - m$.

**Response**

<think> Okay, so I have this problem here: I need to find the maximum possible value of $n - m$ where $m$ and $n$ are positive integers with $m < n$, and the interval $[m, n)$ contains more multiples of 2021 than multiples of 2000. Hmm, interesting. Let me try to break this down step by step.

First, let me understand what exactly is being asked. We have an interval from $m$ to $n$, not including $n$. Within this interval, the number of multiples of 2021 should be greater than the number of multiples of 2000. I need to find the largest possible difference between $n$ and $m$ that satisfies this condition.

[. . . ]

Wait, let's take $m = 2001$, $n = 4000$. But wait, $n$ needs to be larger than $m$, so $m = 2001$, $n = 4001$. Then, in [2001,4001), multiples of 2000: 2000 is less than 2001, so only 4000. So, $B = 1$. Multiples of 2021: 2021*1=2021, 2021*2=4042, which is beyond 4001. So, $A = 1$. Still equal.

Wait, perhaps when $n$ is just after a multiple of 2000, but not too close. Hmm.

Wait, let's try a different $m$. Suppose $m = 2021$, then the interval would be [2021, n). The number of multiples of 2021 in this interval is 1 (if n is beyond 4042) and the number of multiples of 2000 would be the number of multiples of 2000 between 2021 and n. Let's say n = 4042.

Then, multiples of 2000: 2000 is less than 2021, so not included. 4000 is less than 4042, so within [2021,4042). Wait, no, 4000 is less than 4042, so it's still less than n=4042. Wait, but 4000 is less than 4042, so in [2021,4042), multiples of 2000 are 4000 only if 4000 is in the interval. Wait, 4000 is less than 4042, so yes, it's included. Wait, no, 4000 is less than 4042, so [2021,4042) includes 4000? Wait, no, 4000 is less than 4042, so if m=2021, n=4042, the interval [2021,4042) includes 4000? No, because 4000 is less than 2021. So, from 2021 to 4042, the multiples of 2000 are 4000 is not included, next is 6000 which is way above. Wait, actually, the first multiple of 2000 after 2021 is 4000? Wait, 4000 - 2021 = 1979, which is more than 2000, so the next multiple would be 4000 + 2000 = 6000.

[. . . ]

If m=1, n=2000*k +1, n=2000*k + something. Wait, perhaps it's better to think in terms of how many multiples of 2000 and 2021 are included.

Wait, perhaps the maximum n - m occurs when the interval includes as many multiples of 2000 as possible without including any multiples of 2021. Alternatively, the other way around.

Wait, but in the previous example, m=1, n=4043, [1,4043), which includes 2000 and 4000 as multiples of 2000, so A=2, and includes 2021 and 4042 as multiples of 2021, so B=2. Thus, A=B=2, so not satisfying A¿B.

But when n=4042, [1,4042), multiples of 2000 are 2000 and 4000, which are both included, so A=2. Multiples of 2021 are 2021 and 4042, which is excluded, so only 2021 is included, so B=1. Thus, A=2 ¿ B=1, which satisfies the condition.

So, n - m = 4042 -1=4041.

Wait, is that correct? Wait, 4042 -1=4041. So, n=4042, m=1. Then, [1,4042) includes 2000,4000 (A=2) and 2021,4041 (but 4042 is excluded). Wait, actually in [1,4042), the multiples of 2021 are 2021, 4042 is excluded. So, B=1.

[. . . ]

Wait, perhaps m= something else.

Wait, let me try m=2021, n=?

So, [2021, n). How many multiples of 2000 are in there? 2021 is included? 2021 divided by 2000 is 1.0105, so floor(2020/2000)=1, so floor((n-1)/2000)=1 or 2?

Wait, if n is 4042, [2021,4042) includes multiples of 2000: 2000 is less than 2021, so first multiple is 4000, which is in [2021,4042), and 6000 is excluded.

So, A=1.

Multiples of 2021: 2021,4042. 4042 is excluded, so only 2021.
Thus, B=1.

[. . . ]

</think>

Figure 20: Example of Chunk on boundaries in Delethink-RL. The horizontal line specifies the chunk boundary.

