# OpenReview forum: "The Markovian Thinker: Architecture-Agnostic Linear Scaling of Reasoning"
_ICLR.cc/2026/Conference — ICLR 2026 Poster_

### Official Review · Reviewer_pDsN · 2025-10-25

**Soundness:** 3
**Presentation:** 3
**Contribution:** 3
**Rating:** 8
**Confidence:** 4

**Summary:**

It proposes a novel reasoning algorithm that enables large language models (LLMs) to perform extended reasoning with linear compute complexity and constant memory usage, in contrast to the standard Long Chain-of-Thought (LongCoT) approach, which incurs quadratic computational costs due to the self-attention mechanism’s dependence on full context length.

A zero-shot method that applies to off-the-shelf reasoning LLMs (1.5B–30B parameters), enabling them to reason over hundreds of thousands of tokens with no performance loss—and often performance gains—compared to LongCoT.

An RL-based training framework that trains LLMs as native Markovian thinkers. It matches LongCoT’s task performance while using 70% less memory and achieving 40% faster token generation.

**Strengths:**

1.  It introduces a genuinely novel paradigm—Markovian Thinking—for long-horizon reasoning in large language models (LLMs). While prior work has attempted to compress, prune, or summarize reasoning traces to mitigate the quadratic cost of attention, Delethink reframes the problem entirely: it assumes that only a fixed-size suffix of prior reasoning is necessary to continue coherent thought. This is a conceptual shift from “how to shorten thinking” to “how to make thinking Markovian.”
2. Comprehensive empirical evaluation across multiple model scales (1.5B–30B), benchmarks (AIME, GPQA-Diamond, LiveCodeBench), and reasoning lengths (up to 128K tokens).
3. Efficient test-time scaling: Models can reason far beyond their training-time token budgets without retraining.
4. Feasible RL training on long-horizon tasks (e.g., 96K-token reasoning would require ~27 H100-months with LongCoT vs. ~7 with Delethink).

**Weaknesses:**

1. lacks of comparision between mem-agent[1] on long-context-generation task.
2. It carries the last few thousand thinking tokens across chunks enables LLMs to continue their reasoning while matching or surpassing
their LongCoT performance. Did you compare this to summarization of the chunks.



[1] MemAgent: Reshaping Long-Context LLM with Multi-Conv RL-based Memory Agent

**Questions:**

1. How to select the per-chunk context size C of different size of model and tasks?
2. the different between this work and mem-agent[1].
3. Does the different rl algorithm have different effects on preformance such ppo/reinforce++ ?



[1] MemAgent: Reshaping Long-Context LLM with Multi-Conv RL-based Memory Agent

---

> ### Author Response · Authors · 2025-11-14
>
> We thank the reviewer for their thoughtful feedback and positive assessment, especially recognizing the novelty of our paradigm, the comprehensive empirical evaluation, and the efficiency gains that make long-horizon RL training practically feasible.
>
> **Comparison to Mem-Agent**
>
> Mem-Agent is concerned with long-text retrieval/QA tasks where the prompt itself is a very long document, e.g. where the prompt includes the text of many books. Mem-agent proposes processing the prompt segment by segment in light of question. Delethink is concerned with long thinking and keeps the prompt intact. Therefore, they are tackling different challenges. Mem-agent has no benefit in reasoning as the prompt of a reasoning problem is short and Delethink does not benefit Long-context retrieval as Delethink keeps the long prompt intact. This shows up in the benchmarks as well. Mem-agent does not show results on math/code reasoning tasks and evaluates on RULER. That said, both Delethink and Mem-agent are directionally targeting architecture-agnostic linear compute and constant memory in different areas leading to significant progress at scale. They can also be combined to make a general agent that works well on both domains. This is an interesting direction and we leave that to future work.  We'll cite and discuss Mem-agent in the final version of the paper.
>
> **Summarization of Chunks**
>
> > Did you compare this to summarization of the chunks.
>
> Although thought summarization and continuing reasoning based on the summary is a plausible alternative, it’s not a natural behaviour of the current reasoning LLMs. It must be induced by some form of training on synthetic data/supervision [1]. That’s why direct fair comparison to Delethink is challenging. Delethink, on the other hand, can be in principle applied to any reasoning model zero-shot without any prompting or training.
> We also note that naive post-hoc summarization of chunks (e.g., asking another LLM to summarize) can be unreliable. For example, if the chunk is terminated in the middle of solving an equation, the summary must contain enough information to allow the model to continue solving the equation, which is not guaranteed in practice.
>
> **Questions**
>
> > How to select the per-chunk context size C of different size of model and tasks?
>
> The context size C is a hyperparameter of the algorithm that should be tuned for a specific setting. If Delethink is to be applied zero-shot it should be tested on that setting based on tolerable context length and the desired task performance. If RL is possible, Delethink-RL can teach the model to operate on a diverse range of context lengths. In Fig 13, we show training with 8K, 48, and 2K context size and still with even 2K the model is learning effectively. For example, this can be especially useful for edge devices where memory is constrained.
>
> > Does the different rl algorithm have different effects on performance such ppo/reinforce++ ?
>
> Delethink is a change to the thinking environment and in principle can be trained with any RL algorithm. That said, we have not tested the performance of other RL algorithms. One could imagine that PPO due to its finer credit assignment mechanism can differentiate better between advantageous and non-advantageous chunks.
>
> **Scaling to 96K tokens**
>
> As a final note, to demonstrate the scalability of Delethink, we further trained the Delethink-24K checkpoint with a 96K budget (Fig 16 at the end of updated PDF for rebuttal). This increased its AIME’24 accuracy to 49% (52% with test-time scaling) and extended its average thinking length to 35K. In contrast, training an equivalent LongCoT model would exceed our computational budget due to its quadratic training cost.
>
> We appreciate the reviewer’s careful evaluation of our work and hope that our responses have clarified the key points; we are happy to elaborate further if any issues remain.
>
> ---
> [1] Yan, Yuchen, et al. "Inftythink: Breaking the length limits of long-context reasoning in large language models." arXiv preprint arXiv:2503.06692 (2025).

---

> > ### Comment · Reviewer_pDsN · 2025-11-24
> >
> > Thank you for your reply.
> >
> > I believe most of my concerns have been addressed.
> > The only remaining concern is the performance of different reinforcement learning (RL) algorithms within this paradigm. I think it would be valuable to evaluate them across a range of algorithms to help the community identify more efficient approaches—such as the one proposed in [1].
> >
> > I’ll keep my current score.
> >
> > [1] Scaling Long-Horizon LLM Agents via Context-Folding

---

> > > ### Author Response · Authors · 2025-12-01
> > >
> > > We thank reviewer for their suggestion and overall positive assessment of our work.
> > >
> > > Although [1] has appeared on arxiv after the submission deadline, we'll still cite and discuss this in the final version of the paper.
> > >
> > > [1] Scaling Long-Horizon LLM Agents via Context-Folding.

---

### Official Review · Reviewer_YiPw · 2025-11-01

**Soundness:** 3
**Presentation:** 4
**Contribution:** 4
**Rating:** 6
**Confidence:** 4

**Summary:**

The paper deals with efficiency in reasoning language models, particularly the compute cost of long chain-of-thought. The paper proposes Delethink, which generates reasoning in chunks, with each chunk conditioned only on the prompt and a fixed-size subset of tokens from the previous chunk. This achieves more favorable compute scaling and constant memory, while maintaining or modestly decreasing accuracy in the settings tested.

**Strengths:**

- Simple, intuitive, and creative method based on feeding in subsets of the reasoning tokens-so-far.
- The zero-shot finding is surprising: feeding the truncated reasoning trace-so-far can match performance of using the full reasoning trace, without additional training, in some settings (Figure 4).
- Leads to good gains in efficiency (training cost, inference throughput, computational cost of scaling the token budget).
- Enables improved performance at large token budgets in the tested settings.
- Clear formalization of the method and clearly written paper.

**Weaknesses:**

- Most analysis is done on AIME 24 and AIME 25. It's unclear whether the results generalize to other settings, which is particularly noteworthy due to the assumptions present in the method (discussed in the next two points).
- In the zero-shot setting, the method relies on the Markov assumption; concretely, that the relevant context is contained in the beginning and ending of the chunk of the reasoning trace. It is indeed surprising that this assumption held for Qwen3 30B-A3B on the math reasoning datasets in Figure 4. However, I am skeptical that this would hold in many tasks. If so, the zero-shot setting may be of limited use.
- When the model is trained, the learning algorithm should ideally put the relevant context into the beginning/end part of the chunk so that it is available at the next step. I have two concerns:
    - The zero-shot experiments seem to suggest that standard LongCoT may naturally place tokens in positions that are amenable to Delethink. Do you have evidence that the model learns to place tokens much differently than with standard Long CoT training?
     - Can you provide experiments on a broader range of tasks beyond AIME to demonstrate generalization? The CrossWordBench results in Figure 5 suggest potential limitations when task state cannot be compressed into the carried context.
- Delethink underperforms standard long CoT at reasonable thinking budget levels (4k-21k, Figure 8).
- Delethink Training appears to be a standard RL algorithm applied to the MDP defined by the Delethink setting. I'm fine with this if it works well and the paper isn't overclaiming; however I'm listing it as a minor weakness to denote that the training is not particularly surprising or innovative.
- The method is referred to as a "thinking algorithm"; it is unclear what this means.

**Questions:**

Please address the statements and questions described in the Weaknesses above.

---

> ### Author Response · Authors · 2025-11-14
>
> We thank the reviewer for their thoughtful and positive assessment, especially highlighting our method’s simplicity, surprising zero-shot effectiveness, efficiency gains at large token budgets, and clear presentation.
>
> **Evaluation on more benchmarks**
>
> Thanks for the suggestion. We have updated the paper to include evaluations on additional standard reasoning benchmarks (see Fig. 15 at the end of the updated PDF), including HMMT (math), GPQA (Ph.D.-level questions), and LiveCodeBench (coding). These benchmarks exhibit the same pattern of results as AIME’24/’25 and further support the generality of Delethink.
>
> **Broader Tasks and Generality of Zero-shot Settings**
> >In the zero-shot setting, the method relies on..., I am skeptical that this would hold in many tasks. If so, the zero-shot setting may be of limited use..
>
> >Can you provide experiments on a broader range of tasks beyond AIME to demonstrate generalization? The CrossWordBench results in Figure 5 suggest potential limitations when task state cannot be compressed into the carried context.
>
> We evaluate zero-shot Delethink on a diverse set of tasks: math (AIME’24, AIME’25), coding (LiveCodeBench), Ph.D.-level QA (GPQA Diamond), and puzzles (CrossWordBench), to test how broadly the Markov assumption holds. CrossWordBench is deliberately chosen as a challenging case, since solving crosswords requires maintaining a live board state and deletions can remove already solved cells. Zero-shot Delethink is surprisingly robust overall, but, as discussed in Section 3.1 (“Limits of zero-shot Delethink”), it does have limitations on such stateful tasks. We hypothesize that Delethink-RL can teach the model to work effectively with this restricted Markovian state. In the updated paper we show an example of this. In Figure 13 (at the end of updated PDF for rebuttal), we trained a model with only 2K context size. While the initial performance is low, it leans with RL to use that context well.
>
> **Mechanics of Delethink**
> >The zero-shot experiments seem to suggest that standard LongCoT may naturally place tokens in positions that are amenable to Delethink. Do you have evidence that the model learns to place tokens much differently than with standard Long CoT training?
>
> Although we have not performed a systematic large-scale analysis (note that 24K tokens is roughly 60 pages of a book), we inspected multiple representative examples, including Delethink behavior at chunk boundaries (see examples in Figs. 17 and 18), and did not observe clear qualitative differences. That said, such differences may simply not be apparent by visual inspection; the model could encode them implicitly in its token distributions.[1]
>
> **Lower Performance in Short Token Budgets**
> >Delethink underperforms standard long CoT at reasonable thinking budget levels (4k-21k, Figure 8).
>
> On AIME’25, when evaluated under the training token budget of 24K, Deletethink-RL attains slightly lower accuracy than LongCoT-RL at some shorter token lengths: 29.03% vs. 29.11% under a 16K budget, and 25.00% vs. 26.24% under a 10K budget. This suggests that Deletethink-RL learns to actively utilize its available thinking budget of 24K, which can lead to a small accuracy drop when the test-time budget is significantly reduced relative to training(e.g., 24K -> 10K).
>
> Note that these models are trained with access to a full 24K budget. When a lower budget is preferred, Deletethink-RL can instead be trained with this smaller budget, explicitly enforcing the desired constraint. We will add a discussion of this phenomenon and its implications for different token budgets in the revised version of the paper.
>
> **Delethink MDP**
>
> Yes. One could actually view Delethink as a change to the thinking environment MDP where the environment resets the thinking after a chunk context is exceeded and prompts the model with the query and last few thousand thinking tokens. While the LongCoT environment was assumed implicitly as given in the reasoning literature, this simple change in the environment (without changing the architecture and with simple implementation) led to linear thinking and constant memory with no performance loss.
>
> **Scaling to 96K tokens**
>
> As a final note, to demonstrate the scalability of Delethink, we further trained the Delethink-24K checkpoint with a 96K budget (Fig 16 at the end of updated PDF for rebuttal). This increased its AIME’24 accuracy to 49% (52% with test-time scaling) and extended its average thinking length to 35K. In contrast, training an equivalent LongCoT model would exceed our computational budget due to its quadratic training cost.
>
> We are grateful for the thoughtful discussion, which has guided several refinements to our work. We hope our response clarifies the outstanding points and encourages a more positive assessment of our work.
>
> ---
> [1] Cloud, Alex, et al. "Subliminal learning: Language models transmit behavioral traits via hidden signals in data." arXiv preprint arXiv:2507.14805 (2025).

---

> > ### Comment · Reviewer_YiPw · 2025-11-18
> >
> > Thank you for the detailed response and the additional results, which further strengthen the paper. Overall I think the idea is creative and has been thoroughly investigated. I think the paper should be accepted, so I have increased my score to 8.

---

### Official Review · Reviewer_ntQj · 2025-11-01

**Soundness:** 2
**Presentation:** 3
**Contribution:** 2
**Rating:** 4
**Confidence:** 3

**Summary:**

This paper introduces Delethink, a novel algorithm designed to overcome the quadratic computational cost associated with long chain-of-thought (LongCoT) reasoning in Large Language Models (LLMs). The core idea is the "Markovian Thinking Paradigm," where the model reasons in a sequence of short, fixed-context chunks. Each new chunk conditions only on the original prompt and a small suffix (the "Markovian state") of the previous chunk, effectively "deleting" the rest of the reasoning history. This approach decouples the total reasoning length from the computational context, leading to linear scaling of compute and constant memory usage.

**Strengths:**

The Markovian Thinking Paradigm: A new formulation for LLM reasoning that avoids the quadratic cost of self-attention over long sequences.
An RL training framework to create native Markovian thinkers, which is shown to be significantly more compute- and memory-efficient than standard LongCoT training while achieving competitive or superior performance.
Comprehensive experiments demonstrating that Delethink matches or surpasses LongCoT performance on complex reasoning benchmarks (e.g., AIME, GPQA) while being 40% faster and using 70% less memory, and enabling effective test-time scaling beyond training-time limits.

**Weaknesses:**

The authors should analyze the content of the reasoning traces in successful vs. failing cases. What information is being lost when tokens are deleted?
The claims of orthogonality to methods like KV cache eviction and quantization are valid, but a combined evaluation is missing.
If deleting already generated tokens is incompatible with KV cache technology, how does the inference speed compare to LongCoT?
In lines 247-248, "we vary the thinking budget from 8K to 256K tokens." How is the thinking budget controlled?

**Questions:**

See weakness.

---

> ### Author Response · Authors · 2025-11-14
>
> We appreciate the reviewer’s time and feedback. We thank the reviewer for acknowledging the strong performance, effective test-time scaling, and the comprehensive evaluation setup.
>
> **Examples of Delethink Traces**
>
> Thank you for the suggestion. In our qualitative analysis, we do not observe a clear systematic difference between the reasoning traces of successful and failing cases: across chunk boundaries, the model continues its reasoning as usual (see Fig. 17 at the end of the updated PDF for the rebuttal). As discussed in Section 3.1 (Problem Coverage) and Figure 5, almost all questions solvable by LongCoT are also solvable by Delethink. This suggests that there is not much failing due to missing context, but the model might just answer wrong as LongCoT does indicating that it has the information it needs to continue reasoning (note that 8K tokens correspond to roughly 20 pages of a book, which already provides substantial context). Following your suggestion, we have updated the paper with randomly sampled examples of Delethink traces in the appendix to shed further light on this behavior.
>
> **KV Cache**
> > The claims of orthogonality to methods like KV cache eviction and quantization are valid, but a combined evaluation is missing.
>
> We added GPT-OSS to our zero-shot application of Delethink in Fig. 14, showing its effectiveness. GPT-OSS has a hybrid attention architecture where in every two layers it replaces the global attention with sliding window attention (which limits the KV cache)
>
> In practice, more involved approaches require careful GPU kernel engineering in both the inference and training stacks to achieve the desired runtime efficiency. For example, the recent implementation of KV eviction/compression in vLLM (https://github.com/PiotrNawrot/sparse-frontier) currently only supports a batch size of 1, which severely limits its efficiency gains in realistic settings. Precisely because of these challenges, Delethink is deliberately designed to be simple: it delivers linear compute and constant memory without modifying the model architecture, making it straightforward to implement on existing infrastructure. As a final note, despite the general consensus that pure sliding window as KV eviction/compression strategy perform poorly in long-context retrieval [1], Delethink provides evidence that they might work well in reasoning if implemented.
>
> **Delethink vs. LongCoT Inference Speed**
>
> Yes. It is correct that when Delethink continues to the next chunk, the KV cache needs to be recomputed which adds a prefill cost, but this is a constant-factor overhead and the overall time complexity remains linear and the memory remains constant. Note that in Figure 2(C) we are reporting empirically measured generation throughputs under Delethink vs. LongCoT which shows that for thinking 96K tokens Delethink is almost 5X faster than LongCoT.
>
> **Questions:**
> > In lines 247-248, "we vary the thinking budget from 8K to 256K tokens." How is the thinking budget controlled?
>
> For test-time scaling, we right-trimmed each trace so its thinking token count matched the specified test-time budget. We’ll update the paper with this explanation:
>
> **Scaling to 96K tokens**
>
> As a final note, to demonstrate the scalability of Delethink, we further trained the Delethink-24K checkpoint with a 96K budget (Fig 16 at the end of updated PDF for rebuttal). This increased its AIME’24 accuracy to 49% (52% with test-time scaling) and extended its average thinking length to 35K. In contrast, training an equivalent LongCoT model would exceed our computational budget due to its quadratic training cost.
>
>
> We appreciate the reviewer’s comments and hope our response has resolved the raised issues, and encourages a fresh evaluation of our work. We’re happy to address any remaining questions.
>
> ---
> [1] Li, Yucheng, et al. "Scbench: A kv cache-centric analysis of long-context methods." arXiv preprint arXiv:2412.10319 (2024).

---

> ### Author Response · Authors · 2025-11-24
>
> Dear Reviewer,
>
> Thank you again for your dedication to the review process. We wanted to kindly ask whether you’ve had a chance to read our response. We hope it has addressed your raised concerns. Reviewers iAzf and YiPw found the rebuttal and additional results helpful and have subsequently increased their scores to 8. We hope our response also encourage you to give a fresh assessment of our work.

---

> ### Comment · Reviewer_ntQj · 2025-11-25
>
> Thank you for your response — it has alleviated some of my concerns. However, one possible explanation for why removing earlier tokens does not hurt performance is that the model may have a limited ability to utilize long-range context. As models become better at leveraging earlier context (for example, by increasing model size), would removing earlier context lead to larger performance degradation in such cases?
>
> For instance, in Figure 14, the LongCoT performance of GPT-OSS 120B appears much harder to match compared with Qwen3 30B-A3B.

---

> > ### Author Response · Authors · 2025-11-26
> >
> > Thanks for the fruitful discussion initiated.
> >
> >
> > Delethink's effectiveness does not stem from poor long-context ability: these models are already strong on long-context benchmarks (e.g., Qwen3-30B-A3B-Thinking reaches 96.8% on RULER[1] at 128K). Therefore, the effectiveness of zero-shot Delethink shows the Markovian nature of thinking traces. In other words, this is a characteristic of reasoning/thinking, whereas on tasks like RULER or long-context QA, deleting earlier tokens would indeed hurt performance. Even as model scale and long-context capability improve, our results show that the effective thinking state remains much shorter than the full context window (e.g., 16K vs. 128K).
> >
> > > For instance, in Figure 14, the LongCoT performance of GPT-OSS 120B appears much harder to match compared with Qwen3 30B-A3B.
> >
> > Zero-shot Delethink catches up with LongCoT for both GPT-OSS 120B and Qwen3-30B-A3B. Larger models may require a somewhat larger Markovian state, but our paper’s message is that this is still far below the full context length.
> >
> > Besides zero-shot Delethink, once Delethink-RL training starts, the model explicitly learns to reason under this constraint: even under extreme context pressure (C = 2K), Delethink-RL learns effectively (Fig. 13), and we expect this to become even easier with stronger base models.
> >
> > We appreciate the reviewer’s feedback and integrate this discussion in our paper. We hope our response has resolved the raised concern and encourages a more positive evaluation of our work. We’re happy to address any remaining questions.
> >
> > [1] https://huggingface.co/Qwen/Qwen3-30B-A3B-Thinking-2507

---

> > > ### Comment · Reviewer_ntQj · 2025-11-27
> > >
> > > Thank you for your response. I believe the authors’ reply has alleviated my concerns, and I will raise my score.

---

### Official Review · Reviewer_iAzf · 2025-11-03

**Soundness:** 3
**Presentation:** 3
**Contribution:** 3
**Rating:** 6
**Confidence:** 4

**Summary:**

The paper introduces Delethink, an algorithm aimed to mitigate the quadratic computational cost of long chain-of-thought (LongCoT) reasoning in LLMs. The proposed method, replaces a single monolithic reasoning trace with a sequence of shorter chunks. Each chunk is generated by conditioning only on the original prompt and a fixed-size suffix of the preceding chunk, allowing the model to theoretically scale its reasoning length with linear compute and constant memory. The paper presents two primary use cases: a zero-shot inference strategy for off-the-shelf models and an RL training framework (Delethink Training). Empirical results show that Delethink can match the performance of standard LongCoT on reasoning benchmarks while claiming to be more efficient in terms of speed and memory usage.

**Strengths:**

1. The paper addresses the critical and timely problem of controlling increasing complexity in LLM reasoning because of long chain-of-thought.
2. The proposed Delethink algorithm is simple and can be applied as a zero-shot inference strategy to existing models without requiring any kind of architectural modifications.
3. The results are generally comprehensive, showing stronger performance at lower costs for both inference and training of reasoning models.

**Weaknesses:**

1. Identical compute budgets: Major figures in the paper are compared based on ``thinking tokens'' but Delethink also has to recompute kv cache and q values after every chunk. A more fair comparison would have been if wall-clock and optionally TFLOPs across methods would have been reported.
2. From results, it seems that the performance of Deletethink is heavily dependent on the choice of length of chunk context. For instance, 1.5 and 7B models, it performs well with 8K on GPQA, but for AIME it performs better with 4K. Similarly, for Qwen3-30B even 8K performs worse than 16K. Given strong dependence it is unclear how can the optimal length (or even the one that performs better than longCoT) of chunk context be determined.
3. ``Summarization-based reasoning'' (e.g., iterative summarization, InftyThink) enablelonger effective thinking at lower cost, including at least one of these baselines is essential (e.g., iterative summary every K tokens).
4. From figure 2a, it seems that for certain token lengths (at lower token lengths), Deletethink performs worse than LongCoT. Further, since we don't results with wall time on x-axis are not shown, it is unclear whether the same trend holds for other model/dataset combinations as well. If this is generally true, then this should be acknowledged in the limitations section.

The above four points are the primary weaknesses of the paper.

In addition:

## Minor Weaknesses/Suggestions:

5. Ablate the ``fold first 100 tokens of initial chunk into q.'' This is a useful practical tweak but undermines the purely markovian nature claim. Can authors show how much of the zero-shot benefit depends on this?
6. Other types of baselines: Comparison of LongCoT with sliding window (if architecturally possible) or KV eviction, i.e., do not re-prefill q between chunks could strengthen the claims.
7. Suggestions for writing and presentation:
    - The section 4.1 could be simplified in formulation. For instance, variable C is introduced but not used again. Similarly, the need for extending length from n to nS seems unnecessary?
    - There are several empty ablation sections such as A.5, A.6.1, A.6.2.

**Questions:**

Please see Weaknesses Above.

---

> ### Author Response · Authors · 2025-11-14
> **Rebuttal 1/2**
>
> We thank the reviewer for their time and consideration and their overall positive and constructive feedback acknowledging the strong performance and lower cost while being simple and architecture-agnostic.
>
>
> **Comparison of FLOPs and Wall Clock Time.**
>
> Even under 30K, because generation is memory-bound as discussed in Sec. 4.1, the throughput is inversely proportional to context length. This is consistent with our empirical results: Delethink achieves about 40% faster rollouts during RL at 24K budget(Section 6.1), and  for an 96K average thinking length, Delethink rollouts finish at least 4x faster than LongCoT(Figure 2).
>
> To address your request for wall-clock numbers, on AIME’25 with an identical 24K thinking budget Delethink takes 1.81 s per rollout, compared to 2.08 s for LongCoT. However, beyond the training budget of 24K (test-time scaling) wall clock time comparison becomes arbitrary as LongCoT stops almost all of its thinking quickly by emitting <eos> whereas Delethink continues thinking to 128K tokens.
>
> **Sensitivity to Chunk Size**
>
> We agree that, for zero-shot application of Delethink, one may need to optimize the hyperparameter for chunk/markovian state. However, Delethink-RL trains models to work well with any context size. In Figure 13 (at the end of updated PDF for rebuttal), we ablate training with chunk sizes of 2K, 4K, and 8K. The 4K and 8K settings perform very similarly, and even 2K, while somewhat weaker, still yields clear gains with Delethink-RL (such a small chunk size enables deployment on edge devices).
>
> Moreover, Delethink-infer consistently achieves high coverage of positive samples across chunk sizes from 4K to 16K on various models (R1-distill family, Qwen3, and GPT-OSS 120B which we added in rebuttal). This provides a strong initialization for RL to pick up and then reinforce the Markovian behavior in the desired chunk size as the RL training starts with many positive samples.
>
>
> **Comparison To InftyThink**
>
> The AIME’24 accuracy reported in the InftyThink paper[1] for the same base model (26.04%) is lower than that of Delethink-RL (40.1%, and 46.0% with test-time scaling up to 128K tokens).
>
> To the best of our knowledge, Delethink-RL is the first method to achieve linear-time scaling of RL training for reasoning, whereas InftyThink does not employ RL. InftyThink’s summarization behavior is not a learned policy or zero-shot behavior, but is distilled from synthetic data constructed by combining outputs from several models through SFT. Therefore, as InftyThink is not RL based, it cannot be compared to Delethink-RL and as it is not a zero-shot method and requires an involved distillation step, it cannot be compared with zero-shot Delethink. Delethink does not require any synthetic data and, in principle, can be readily applied to any reasoning model.
>
> We agree that applying RL to iterative summarization such as InftyThink is an interesting direction. However, it would require meticulous design and tuning of the RL environment by itself, which we leave for future work.
>
>
> **Lower Performance on certain token lengths**
>
> On AIME’25, when evaluated under the training token budget of 24K, Deletethink-RL attains slightly lower accuracy than LongCoT-RL at some shorter token lengths: 29.03% vs. 29.11% under a 16K budget, and 25.00% vs. 26.24% under a 10K budget. This suggests that Deletethink-RL learns to actively utilize its available thinking budget of 24K, which can lead to a small accuracy drop when the test-time budget is significantly reduced relative to training(e.g., 24K -> 10K).
>
> Note that these models are trained with access to a full 24K budget. When a lower budget is preferred, Deletethink-RL can instead be trained with this smaller budget, explicitly enforcing the desired constraint. We will add a discussion of this phenomenon and its implications for different token budgets in the revised version of the paper.
>
> **Please refer to the next reply for responses to the rest of questions due to character limits.**
>
> [1] Yan, Yuchen, et al. "Inftythink: Breaking the length limits of long-context reasoning in large language models." arXiv preprint arXiv:2503.06692 (2025).

---

> ### Author Response · Authors · 2025-11-14
> **Rebuttal 2/2**
>
> ...second part of the rebuttal
>
> **Questions:**
>
> > Ablate the ``fold first 100 tokens of initial chunk into q.
>
> We thank the reviewer for suggesting this as this strengthened the paper. In response, we ran an ablation of the “folding first 100 tokens” in the zero-shot Delethink setting with R1-Distill 1.5B on AIME’24. Disabling this folding reduces performance from 37.2% to 31.9%. It seems without folding 100 tokens, the thinking starts from the middle of reasoning, the prompt becomes too OOD.  We will include this ablation and its discussion in the final version of the paper.
>
> > KV-cache eviction comparision
>
> Yes, although different from Delethink, in principle one could instantiate a Markovian thinking RL paradigm using a sliding window with the prompt pinned in context, or via KV eviction schemes simulating constant memory. To the best of our knowledge, however, such RL baselines have not yet been realized.
>
> In practice, these approaches require careful GPU kernel engineering in both the inference and training stacks to achieve the desired runtime efficiency. For example, the recent implementation of KV eviction/compression in vLLM (https://github.com/PiotrNawrot/sparse-frontier) currently only supports a batch size of 1, which severely limits its efficiency gains in realistic settings. Precisely because of these challenges, Delethink is deliberately designed to be simple: it delivers linear compute and constant memory without modifying the model architecture, making it straightforward to implement on existing infrastructure. As a final note, despite the general consensus that sliding window as KV eviction/compression strategy perform poorly in long-context retrieval  [2], Delethink provides evidence that they might work well in reasoning if implemented.
>
> > Writing and Presentation
>
> Thanks for catching these. We’ll fix it in the final version.
>
> **Scaling to 96K tokens**
>
> As a final note, to demonstrate the scalability of Delethink, we further trained the Delethink-24K checkpoint with a 96K budget (Fig 16 at the end of updated PDF for rebuttal). This increased its AIME’24 accuracy to 49% (52% with test-time scaling) and extended its average thinking length to 35K. In contrast, training an equivalent LongCoT model would exceed our computational budget due to its quadratic training cost.
>
> Thank again for the fruitful discussion initiated here. it has been valuable in refining our work. We’re happy to address any remaining concerns and hope our response has contributed to a more positive reassessment of the work.
>
> ---
> [2] Li, Yucheng, et al. "Scbench: A kv cache-centric analysis of long-context methods." arXiv preprint arXiv:2412.10319 (2024).

---

> > ### Comment · Reviewer_iAzf · 2025-11-15
> >
> > I thank the authors for their comprehensive response and new results (which are very strong!). I still think that training with summaries instead of purely Markovian view would yield stronger results, but this is an opinion and potential future work rather than a weakness. Therefore, I have increased my score!

---

### Author Response · Authors · 2025-12-01
**Summary of the Rebuttal**

We thank all reviewers for their time and constructive feedback. During the rebuttal phase, we updated the draft with additional ablations (different context sizes), new scaling results (96K), more qualitative examples, and results on larger models (GPT-OSS 120B); and addressed the main concerns and questions raised.

As a result of the discussion, **3 out of 4 reviewers increased their scores, moving from 8664 (avg = 6.0) to 8886 (avg = 7.5)**. Note that *all* these updates are explicitly stated by reviewers which can be seen in the discussion below. In particular:
- Reviewer iAzf: **6 → 8**
- Reviewer ntQj: **4 → 6**
- Reviewer YiPw: **6 → 8**
- Reviewer pDsN: **remained at 8**

---

### Meta-Review · Area_Chair_himH · 2026-01-06

**Summary:**

This is a timely paper that provides a method to reduce reasoning computation & memory cost, to deliver stronger result with lower cost. Both the authors and reviewers engaged in the discussion with major concerns resolved. There is a consensus after the rebuttal that this paper contributes meaningfully to the field and should be accepted. (8886).

**Reviewer Concerns:**

- Fair compute comparison: resolved with wallclock numbers.
- Sensitivity to Chunk Size: extra ablation provided.
- Other alternative ideas: Alternative methods like summarize the context / context folding are proposed by reviewers and not resolved, but the reviewers agree that it should be future work to compare among them.

**Reviewer Scores:**

Reviewers participated in the discussion, the concerns are resolved and scores increased.

---

### Decision · Program_Chairs · 2026-01-26

Accept (Poster)